# Tumor-intrinsic expression of the autophagy gene Atg16l1 suppresses anti-tumor immunity in colorectal cancer

Lucia Taraborrelli[1,14], Yasin Şenbabaoğlu[2,14], Lifen Wang[1,14], Junghyun Lim[1], Kerrigan Blake[1], Noelyn Kljavin[3], Sarah Gierke[4,5], Alexis Scherl[5], James Ziai[5], Erin McNamara[6], Mark Owyong[6], Shilpa Rao[2], Aslihan Karabacak Calviello[2], Daniel Oreper[2], Suchit Jhunjhunwala[2], Guillem Argiles[7], Johanna Bendell[8], Tae Won Kim[9], Fortunato Ciardiello[10], Matthew J. Wongchenko[11], Frederic J. de Sauvage[3], Felipe de Sousa e Melo[12], Yibing Yan[11], Nathaniel R. West[1,15] ✉ & Aditya Murthy[1,13,15] ✉

Microsatellite-stable colorectal cancer (MSS-CRC) is highly refractory to immunotherapy. Understanding tumor-intrinsic determinants of immunotherapy resistance is critical to improve MSS-CRC patient outcomes. Here, we demonstrate that high tumor expression of the core autophagy gene *ATG16L1* is associated with poor clinical response to anti-PD-L1 therapy in KRAS-mutant tumors from IMblaze370 (NCT02788279), a large phase III clinical trial of atezolizumab (anti-PD-L1) in advanced metastatic MSS-CRC. Deletion of *Atg16l1* in engineered murine colon cancer organoids inhibits tumor growth in primary (colon) and metastatic (liver and lung) niches in syngeneic female hosts, primarily due to increased sensitivity to IFN-γ-mediated immune pressure. ATG16L1 deficiency enhances programmed cell death of colon cancer organoids induced by IFN-γ and TNF, thus increasing their sensitivity to host immunity. In parallel, ATG16L1 deficiency reduces tumor stem-like populations in vivo independently of adaptive immune pressure. This work reveals autophagy as a clinically relevant mechanism of immune evasion and tumor fitness in MSS-CRC and provides a rationale for autophagy inhibition to boost immunotherapy responses in the clinic.

In recent years, immune checkpoint inhibitors (ICIs) such as monoclonal antibodies against PD-1, PD-L1, and CTLA-4 have proven to be clinically efficacious in many cancer types[1]. However, only a small subset of CRC patients has thus far demonstrated responses to ICIs, specifically those exhibiting elevated tumor mutation burden caused by defective mismatch repair machinery (dMMR; also termed microsatellite instability, MSI)[2–4]. The vast majority of CRC patients exhibit MMR-proficient, microsatellite stable (MSS) disease, a subset that is highly resistant to immunotherapy[5–7]. Thus, revealing the determinants of ICI resistance in MSS-CRC is

essential for unlocking anti-tumor immunity via novel therapeutic combinations.

Genome-wide association studies (GWAS) of inflammatory bowel disease (IBD) have revealed key pathways associated with increased intestinal inflammation[8,9]. Exploiting this growing understanding of IBD genetics provides an opportunity to leverage validated immunoregulatory mechanisms and enhance immunotherapy responses in CRC. Among these, a missense variant in the core autophagy gene ATG16L1 (T300A), that regulates its caspase-mediated degradation, is established as a highly significant risk allele for Crohn's disease

(CD)[10–13] as well as a putative prognostic factor in CRC[14] and gastric cancer[15]. ATG16L1 is a key component of the E3-ligase-like autophagosome elongation complex (termed the ATG16L1 complex) required for the catabolic process of autophagy[16]. Here, intracellular cargo including aged or damaged organelles, protein aggregates, and intracellular pathogens is engulfed by the autophagosome and targeted for lysosomal degradation[17–19]. Autophagy plays an essential role in intestinal epithelial cell fitness (e.g., Paneth and stem cells)[20–23]. Furthermore, autophagy regulates innate and adaptive immunity via diverse mechanisms including antigen presentation, effector T cell function and regulation of programmed cell death[24–30]. Recent studies using tumor cell line models have revealed members of the core autophagy pathway as tumor-intrinsic or extrinsic (i.e., host-derived) determinants of immunosuppression[31,32], but their translational relevance remains unknown.

In this study, we investigated IMblaze370, a phase III immunotherapy trial of previously treated MSS-CRC, and identify high *ATG16L1* expression as predictive of poor ICI efficacy. Elevated epithelial *ATG16L1* expression in KRAS mutant MSS-CRC tumors strongly associated with decreased overall survival in response to atezolizumab (anti-PD-L1) alone or in combination with the MEK inhibitor cobimetinib. Importantly, this association was not observed in patients treated with regorafenib alone, highlighting *ATG16L1* expression as an immunotherapy-specific predictive biomarker in IMblaze370. Loss of Atg16l1 in genetically engineered murine CRC organoids attenuated CRC growth in metastatic (i.e., liver, lung) as well as primary (colon) tissue niches. Single cell RNA sequencing of tumors and intratumoral myeloid cells revealed tumor-intrinsic and -extrinsic consequences of Atg16l1 deletion in CRC. ATG16L1 promoted stemness-associated transcriptional programs in the tumor and suppressed inflammation within the tumor microenvironment (TME). Mechanistically, ATG16L1 suppressed type II IFN responses in CRC organoids, and treatment of ATG16L1-deficient cells with a combination of IFNγ and TNF significantly enhanced programmed cell death compared to control CRC organoids. Thus, epithelial ATG16L1 promotes disease progression by increasing resistance to immune pressure and maintaining the stem cell pool in MSS-CRC.

## Results

### Elevated *ATG16L1* expression predicts poor immunotherapy response in CRC patients harboring oncogenic KRAS mutations

Non-MSI CRC is generally considered to be highly resistant to immunotherapy, but it is unclear whether identifiable patient subsets exist with heightened therapeutic sensitivity. To evaluate whether *ATG16L1* expression is associated with response to ICI in non-MSI CRC, we evaluated tumor gene expression data from IMblaze370 (NCT02788279), a large ($n = 363$) multi-center phase III trial in locally advanced or metastatic CRC where disease progression was observed in at least two previous lines of chemotherapy[33]. Anti-PD-L1 monotherapy (atezolizumab) was compared to a combination of anti-PD-L1 with the MAP kinase pathway (MEK) inhibitor cobimetinib (atezolizumab+cobimetinib). Monotherapy with regorafenib, an approved multikinase inhibitor[34,35], was used as a standard-of-care control arm. Importantly, the availability of MSI-status and KRAS genotypes enabled us to refine our investigation and provide a clearer context of the role of ATG16L1 in defined subsets of CRC.

Analysis of *ATG16L1* transcript levels in non-MSI-high tumors demonstrated that elevated *ATG16L1* expression was associated with poor overall survival in both the atezolizumab and the atezolizumab + cobimetinib combination arms in the KRAS mutant, but not in the KRAS wildtype setting (Fig. 1a, left and middle panels). In contrast, *ATG16L1* transcript levels did not associate with differential outcome in the regorafenib arm (Fig. 1a, right panel). *ATG16L1* expression showed a positive correlation with tumor epithelial signatures and negative correlations with immune and stromal signatures[36,37], suggesting that the primary source of *ATG16L1* expression was tumor cells (Fig. 1b). Histopathological evaluation of tumor samples from IMblaze370 demonstrated elevated ATG16L1 protein levels primarily in the tumor epithelium, with minor localization to the tumor stroma (Fig. 1c, dotted lines denote epithelial border; asterisk in lower panel denotes stromal component). Single-cell gene expression analysis of human CRC tumors (GSE146771)[38] confirmed elevated *ATG16L1* transcript levels in the tumor epithelium when compared to other cellular compartments of the CRC tumor microenvironment (Fig. 1d, expression of lineage markers associated with each compartment; Fig. 1e, *ATG16L1* transcript levels in each depicted compartment). Gene expression analysis of patient samples from The Cancer Genome Atlas (TCGA) also showed *ATG16L1* transcript levels to be elevated in human CRC compared to normal adjacent tissue (Supplementary Fig. 1a). Consistent with our findings in IMblaze370, correlational analysis of CRC samples from an independent report (GSE17536)[39] demonstrated association of *ATG16L1* transcript specifically with an epithelial gene expression signature (Supplementary Fig. 1b). Lastly, *ATG16L1* transcript levels were not significantly different across stages, ruling out stage-specific *ATG16L1* expression as a confounding factor for clinical outcome (Supplementary Fig. 1c). Expanded analysis of core components of the autophagosome elongation machinery (illustrated in Supplementary Fig. 2a) showed *ATG16L1* to be preferentially enriched in the tumor epithelium when compared to *ATG3, ATG4B, ATG5, ATG7, ATG10* and *ATG12. ATG4A, ATG4C, and ATG4D* were not detected, likely due to poor transcript coverage or low expression (Supplementary Fig. 2b). Additionally, only *ATG16L1* showed a significant association with poor outcome under immunotherapy regimens in KRAS-mutant disease in IMblaze370 (atezolizumab monotherapy, atezolizumab + cobimetinib; Supplementary Table 1).

Given that IMblaze370 remains the only well-powered clinical investigation for immunotherapy in late stage MSS-CRC, additional analysis was limited to observational studies with significantly smaller cohort sizes of stage IV disease (GSE17536, GSE39582, TCGA; Supplementary Table 2). Immunotherapy was not available as a treatment arm in these studies, and subcohorts of non-MSI patients harboring KRAS mutations further decreased patient numbers. With these caveats, a poor prognostic association was observed for *ATG16L1* in GSE17536, and for *ATG7, ATG10,* and *ATG4A* in GSE39582. Small patient numbers ($n < 25$) precluded analysis of a number of additional datasets (GSE39084, GSE17537, GSE33113, GSE24551, GSE13067, GSE13294, GSE18088, GSE26682, GSE41258, and GSE14333). These analyses suggest a weak or absent association between *ATG16L1* transcript levels and patient outcome in non-immunotherapy settings. Caution must be exercised when considering these studies due to small cohort sizes and lack of appropriate treatment arms.

We next investigated whether KRAS status was associated with differences in the transcriptional profiles of *ATG16L1*-low versus -high tumors. Unbiased differential expression and GSEA (gene set enrichment analysis) between *ATG16L1*-low and *ATG16L1*-high tumors in the KRAS mutant and wildtype settings showed strong concordance, suggesting that KRAS genotypes did not contribute to global transcriptional changes in *ATG16L1*-low tumors (Supplementary Fig. 3). Despite the general similarity between KRAS mutant and wildtype tumors, immune cell deconvolution analysis using CIBERSORT[40] revealed that *ATG16L1*-low tumors showed substantially stronger evidence of increased T and NK cell infiltration in the KRAS mutant setting (Fig. 1f, arrows). Thus, improved survival of immunotherapy-treated *ATG16L1*-low patients in the KRAS mutant setting was associated with increased T and NK cell infiltration. Taken together, we conclude that in non-MSI CRC, patients with *ATG16L1*-low tumors may generate a more inflammatory microenvironment, thereby promoting a productive anti-tumor immune response upon checkpoint inhibition with atezolizumab.

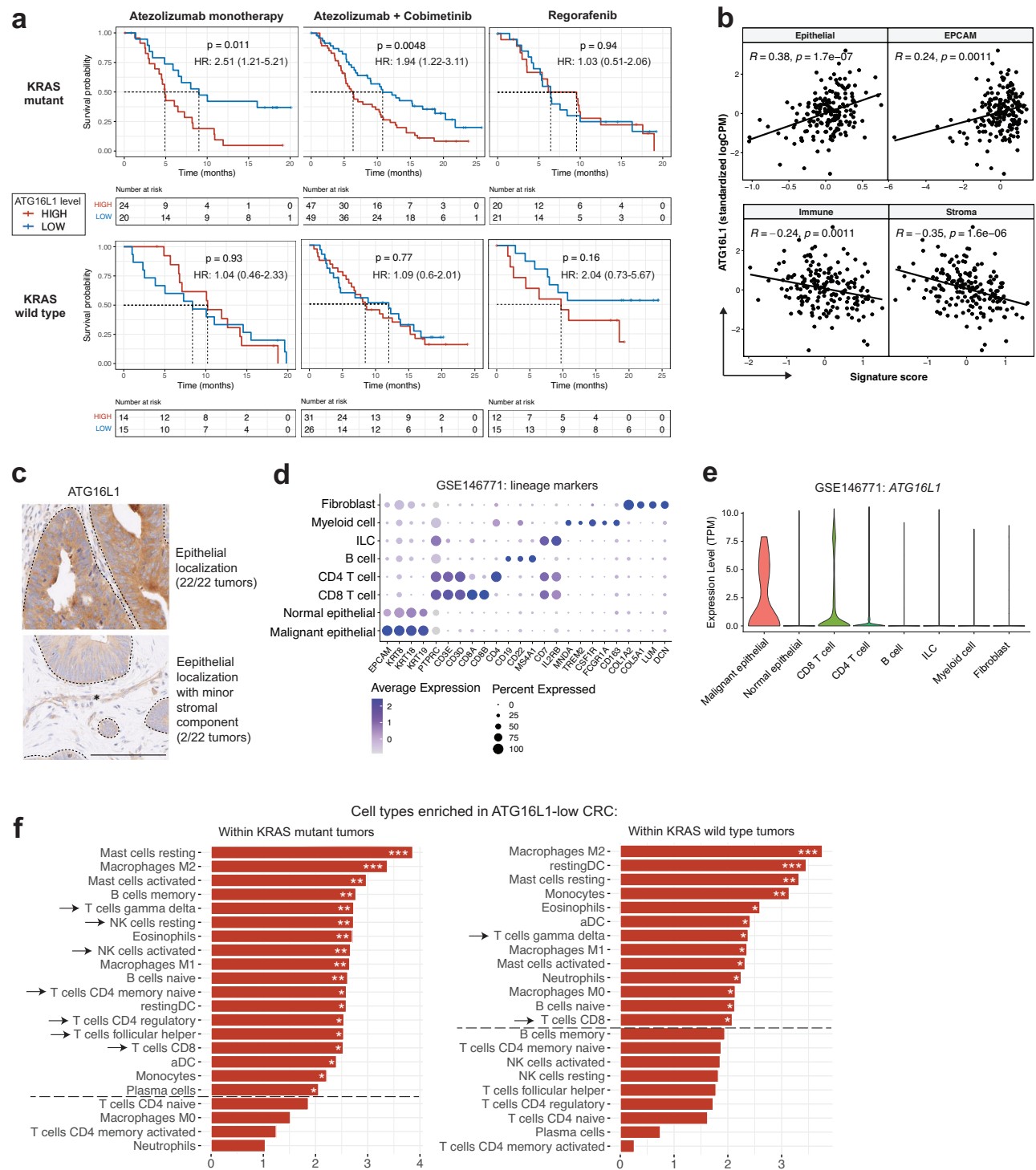

## Tumor-intrinsic ATG16L1 promotes resistance to cellular immunity in vivo

Prompted by the observation that elevated *ATG16L1* expression associates with poor outcome in late-stage CRC, we functionally evaluated its role in CRC progression. First, CRISPR-Cas9 editing of murine colon organoids was used to develop a tumor model harboring key driver mutations observed in MSS-CRC. Loss-of-function in tumor suppressors *Apc*, *Trp53*, *Smad4*, and oncogenic gain-of-function in *Kras (G12D)* were sequentially introduced to develop transformed CRC organoids (termed AKPS, depicted in Fig. 2a). Whole-exome sequencing was performed to validate introduced

mutations (Supplementary Fig. 4a). Next, Atg16l1 was deleted in AKPS CRC organoids and several single cell clones were generated (Supplementary Fig. 4b). ATG16L1 drives autophagosome elongation via lipidation of the ubiquitin-like proteins of the LC3/ATG8 family[16]. Complete loss of lipidated LC3b (LC3-II) was confirmed in Atg16l1 knockout CRC organoid clones (Fig. 2b). Accumulation of a class of proteins called sequestosome-like receptors (SLRs) is a hallmark of defective autophagic flux and Atg16l1 deletion[41]. Consistent with loss of LC3-II, elevated levels of SLRs SQSTM1/p62 and CALCOCO1 were confirmed in Atg16l1 knockout (KO) compared to control (WT) CRC organoids (Fig. 2b).

**Fig. 1 | Elevated *ATG16L1* expression associates with poor outcome in immunotherapy of non-MSI-high CRC harboring oncogenic KRAS mutations.** **a** Kaplan–Meier curves indicating the association between *ATG16L1* transcript levels and overall survival in atezolizumab and regorafenib treated patients (IMblaze370). Median cutoff was used to determine high and low levels separately within KRAS mutant and KRAS wildtype tumors. *P*-values were obtained from log-rank tests. Log-rank hazard ratios (HR) are provided with 95% confidence intervals in parentheses. **b** Scatter plots for KRAS mutant tumors showing the correlation between *ATG16L1* transcript levels and tumor microenvironment signatures for immune, stroma and epithelial cells. Pearson correlation coefficients and two-sided *t*-test *P*-values are shown (*n* = 181 samples). **c** Immunohistochemical (IHC) staining for ATG16L1 protein in tumor biopsies obtained from IMblaze370. Dotted lines indicate tumor epithelial margins, asterisk depicts stromal component. 22 tumor biopsies were analyzed; representative micrographs are shown. Scale bar = 100μm. **d**, Expression of depicted lineage markers (*x*-axis) in major cellular compartments (y-axis) of CRC tumor tissue, analyzed by single-cell RNA sequencing (GSE146771). **e** Comparison of *ATG16L1* transcript levels in each cellular compartment analyzed in (**d**). **f** Immune cell subsets enriched in ATG16L1-low tumors within IMblaze370, as determined by CIBERSORT gene signatures. Arrows indicate T and NK cell subsets. Unadjusted *P*-values from two-sided *t*-tests are shown (KRAS mutant *n* = 181, KRAS wildtype *n* = 113 samples). Dashed lines denote significance threshold at *P* < 0.05. All analysis restricted to non-MSI-high tumors. *$P$ < 0.05, **$P$ < 0.01, and ***$P$ < 0.001. IMblaze370 RNAseq data have been deposited to the EGA under accession number EGAS00001005952, and GSE146771 is publicly available from GEO (see Data Availability section of Methods). Source data (including exact *P*-values) for panel (**f**) are provided as a Source Data file.

CRC metastasis occurs predominantly in the liver and lung[42]. IMblaze370 investigated anti-tumor immunity in the context of advanced disease with pre-existing metastases; we therefore delivered CRC organoids directly to the liver and lung using a modified hydrodynamic tail vein (HTV) injection protocol[43] and conventional intravenous tail vein injection, respectively. Orthotopic injection of CRC organoids into the colonic epithelium was used to establish tumors in the primary disease niche. Tumor growth in the liver and lung models was followed via bioluminescence (BLI) imaging of a stably expressed luciferase reporter. Loss of Atg16l1 drastically reduced liver tumor burden in fully immunocompetent hosts (Fig. 2c, d, i). This finding was confirmed in multiple independent clones lacking Atg16l1 (Supplementary Fig. 5a). Livers from mice administered WT CRC organoids displayed extensive disease burden 6 weeks after inoculation, whereas a minority of mice (<40%) administered Atg16l1 KO CRC organoids presented with only occasional tumor foci (Fig. 2i and Supplementary Fig. 5b–e). Histological analysis showed a significantly decreased tumor area in the Atg16l1 KO group (Supplementary Fig. 5d, e). Stable re-expression of ATG16L1 restored autophagic flux (Supplementary Fig. 5f) and liver growth of KO CRC organoids (Supplementary Fig. 5g–i), demonstrating a direct role of ATG16L1 in promoting CRC fitness. As observed with liver colonization, loss of Atg16l1 significantly decreased the ability of CRC organoids to colonize lung tissue (Supplementary Fig. 6). Lastly, orthotopic implantation of CRC organoids demonstrated that ATG16L1 was also required for tumor growth in the colonic mucosa (Supplementary Fig. 7a–c). Together, our data demonstrate that ATG16L1 promotes CRC growth in immunocompetent mice independent of the tissue niche. As IMblaze370 investigated patient outcomes in post-metastatic CRC, we focused our subsequent experiments on liver colonization of CRC organoids.

Loss of Atg16l1 did not impact basal growth of AKPS organoids in vitro (Supplementary Fig. 4c), suggesting that tumor-intrinsic loss of Atg16l1 may elicit tumor-extrinsic mechanisms of disease control in vivo. This, along with the improved immunotherapy response (Fig. 1a) and elevated T/NK cell-associated gene signatures (Fig. 1f) in *ATG16L1*-low patient subsets prompted us to evaluate the impact of cellular immunity on CRC growth in vivo. We thus implanted WT and Atg16l1 KO CRC organoids into livers of immunocompromised hosts (Fig. 2e–g, summarized in Fig. 2h). First, we asked whether a complete loss of host cellular immunity would accelerate growth of ATG16L1-deficient CRC in the liver. NOD/SCID-gamma IL2Rg^null (NSG) mice are commonly used to investigate growth of human tumor cells in vivo. These mice lack mature B, T, and NK cells, representing a severely immunocompromised host microenvironment. Consistent with a role of cellular immunity in controlling tumor growth, WT CRC organoids grew faster in NSG hosts compared to immunocompetent BL6 mice. Remarkably, Atg16l1 KO CRC organoids grew rapidly in NSG hosts (Fig. 2e, h; WT host *vs* NSG host), to the same extent as WT CRC organoids in control immunocompetent BL6 hosts (Fig. 2d). Although tumor burden in the Atg16l1 KO group remained significantly decreased compared to the WT group (approximately a 2-fold difference in BLI signal after 4 weeks), all NSG mice administered with Atg16l1 KO CRC organoids presented with large tumor nodules (Fig. 2j).

To refine these observations further, we individually depleted host cytotoxic T lymphocytes (CTL, CD8+ T cells) or natural killer (NK) cells followed by implantation of CRC organoids. Near-complete, sustained loss of CD8+ T cells or NK cells was observed 4 weeks following administration of depleting antibodies compared to non-depleting isotype controls (Supplementary Fig. 8a, b). Interestingly, depletion of NK cells markedly rescued growth of ATG16L1 KO CRC organoids, while depletion of CD8+ T cells had no significant effect (Fig. 2f, h; WT host vs. CD8 or NK depleted hosts; tumor growth curves in Supplementary Fig. 8c). Thus, NK cells seem to be key contributors to the clearance of ATG16L1-deficient MSS-CRC.

We next used *Ifng* KO mice to ask whether IFNγ, a critical cytokine that drives effector function of cytotoxic T and NK cells, was involved in tumor control. Compared to WT control hosts (Fig. 2d), loss of host IFNγ significantly increased liver colonization by Atg16l1 KO CRC organoids as shown by BLI quantification (Fig. 2g, h; Supplementary Fig. 8d; WT host vs. IFNγ KO host), macroscopic tumor nodule count (Supplementary Fig. 8e), and histologic examination (Supplementary Fig. 8f, g), suggesting that host IFNγ suppresses the growth of ATG16L1-deficient CRC organoids in vivo.

Overall, we observed improved growth of ATG16L1-deficient CRC organoids in the liver of immunocompromised hosts (Fig. 2h). Consistent with these findings, growth of ATG16L1-deficient CRC organoids was comparable to that of WT control organoids when implanted orthotopically in the colon mucosa of NSG hosts (Supplementary Fig. 7d, e), in contrast to the defective growth of Atg16l1 KO organoids in immunocompetent BL6 hosts (Supplementary Fig. 7b, c). Collectively, our results show that independent of tissue niche, tumor-intrinsic ATG16L1 profoundly limits immune-mediated control of CRC, whereas it has a relatively minor impact when tumors are grown in immunodeficient hosts.

## Loss of Atg16l1 alters the composition and phenotype of CRC organoids

To better understand how ATG16L1 impacted phenotypic programming of cells in the tumor and its microenvironment, we performed transcriptomic profiling of tumors implanted in the livers of NSG mice, since optimal growth of Atg16l1 KO CRC organoids required immunodeficient hosts. Tumors were harvested and live cells were sorted into CRC organoid (eGFP+CD45−) and leukocyte (eGFP−CD45+) fractions. Sorted fractions for each group (WT and Atg16l1 KO, *n* = 2 mice for each) were analyzed by single-cell RNA-sequencing (10x Genomics 3′ scRNA-seq). Data from organoid and immune cells were preprocessed and normalized separately (see Methods). Organoid and immune cell pools were filtered to only keep epithelial and myeloid cells, respectively.

We first focused on tumor-intrinsic states, where unsupervised graph-based clustering revealed eight clusters for epithelial CRC cells

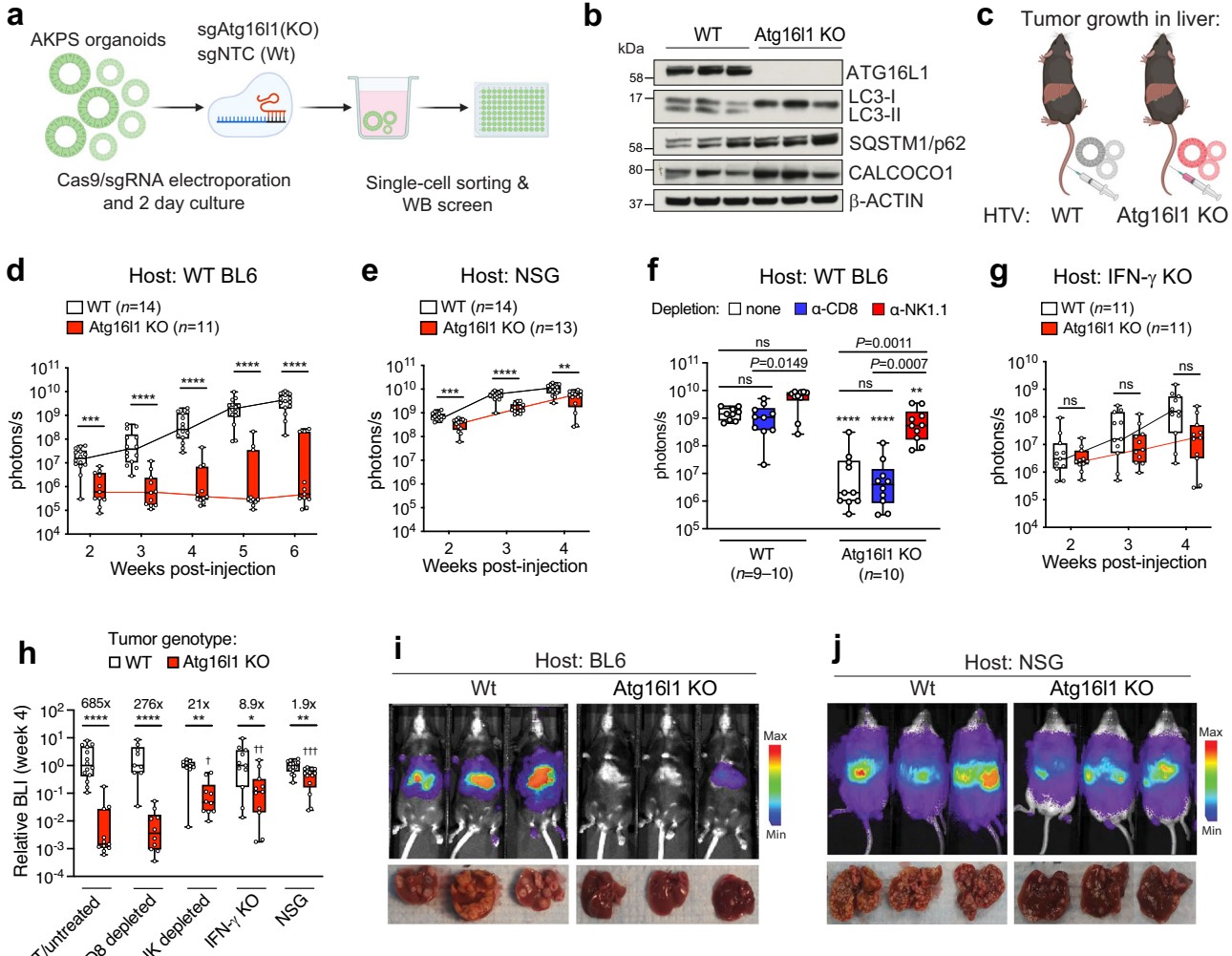

**Fig. 2 | ATG16L1 drives CRC growth in the liver by promoting resistance to cellular immunity. a** Schematic of CRISPR engineering and generation of CRC organoids. **b** Immunoblot analysis of the indicated proteins in WT or Atg16l1 KO CRC organoids Three independent clones of each genotype are shown. **c** Hydrodynamic tail vein (HTV) injection of CRC organoids for liver growth model. **d–h** Bioluminescence imaging (BLI) signal quantification from CRC organoids implanted in the livers of immunodeficient hosts. **d** C57BL/6 J (BL6) immunocompetent hosts. ***$P = 0.0001$, ****$P < 0.0001$. **e** NOD-SCID/Gamma (NSG) immunodeficient hosts; WT control BL6 hosts shown in panel (**d**). **$P = 0.0027$, ***$P = 0.0001$, ****$P < 0.0001$. **f** CD8+ T cell or NK cell depletion in BL6 immunocompetent hosts. Week 5 BLI data are shown. Within tumor genotypes, treatment groups are compared using Kruskal-Wallis test with Dunn's multiple comparisons tests. Atg16l1 KO groups are also compared to their corresponding WT groups using two-sided Mann-Whitney tests; **$P = 0.0021$ and ****$P < 0.0001$. For all Atg16l1 KO groups, $n = 10$ per condition. For WT groups, $n = 9$ for isotype control-treated mice, and $n = 10$ each for anti-CD8 and anti-NK1.1 treated mice. **g** IFN-γ KO hosts; WT control BL6 hosts shown in panel (**d**). ns, not significant. **h** Direct comparison across studies of BLI signal (from panels **d–g**) re-plotted as values normalized to the

medians of WT organoid groups (after 4 weeks of growth). For WT/untreated hosts, $n = 14$ (WT organoids) and $n = 11$ (KO organoids), ****$P < 0.0001$; for CD8 T cell depleted hosts, $n = 10$ per group, ****$P < 0.0001$; for NK-depleted hosts, $n = 10$ per group, **$P = 0.0015$; for IFN-γ KO hosts, $n = 11$ per group, *$P = 0.0233$; for NSG hosts, $n = 14$ (WT organoids) and $n = 13$ (KO organoids), **$P = 0.0027$. WT and Atg16l1 KO organoids were compared in each condition using two-sided Mann–Whitney tests. Atg16l1 KO groups from each immunodeficient condition were also compared to Atg16l1 KO tumor growth in WT/untreated hosts using Mann-Whitney tests; †$P = 0.0048$, ††$P = 0.0032$, †††$P < 0.0001$. **i, j** Representative BLI images of mice (top) and tumor burden in livers (bottom) from (**i**) BL6 ($n = 14$ WT; $n = 11$ Atg16l1 KO) hosts or (**j**) NSG ($n = 14$ WT; $n = 13$ Atg16l1 KO) hosts. In panels (**d–h**), lower and upper hinges in box plots correspond to first and third quartiles, while whiskers extend to minima and maxima. Individual data points indicate separate mice (biological replicates). In panels (**d, e,** and **g**), groups were compared using two-sided Mann–Whitney tests, with $P$-values adjusted for multiple comparisons using the Holm-Sidak method. All data are representative of 2-3 independent experiments. Source data are provided as a Source Data file.

($n = 15,263$): Proliferative (Prolif.), Secretory/sensory (SS), Mature enterocyte (ME), Neuroendocrine (NE), Enterocyte progenitor (EP), Interferon response (IFN resp.), Stem^HI, and Goblet/Paneth (GP) cell clusters (Fig. 3a, b). Stem^HI cluster markers were also expressed in the GP cell cluster; however, unsupervised topic modeling and RNA velocity analysis confirmed the Stem^HI cluster as a distinct pool having a stem cell-related transcriptional program and giving rise to multiple other CRC subpopulations (Supplementary Fig. 9a–f). Labels for other clusters were validated with a correlation analysis comparing cluster

markers with previously established signatures of normal intestinal epithelia[44,45] (Supplementary Fig. 9g).

Comparison of cell type proportions indicated that Atg16l1 deletion substantially altered the composition of CRC organoid cells in vivo (Fig. 3c–e). In the Atg16l1 KO group, there was a significant increase in the proportion of IFN responsive CRC cells, pointing to tumor-intrinsic immunosuppressive processes dependent on autophagy (Fig. 3e). The Atg16l1 KO group also exhibited a proportional increase in the NE and SS response clusters (Fig. 3e). In contrast, there was a significant

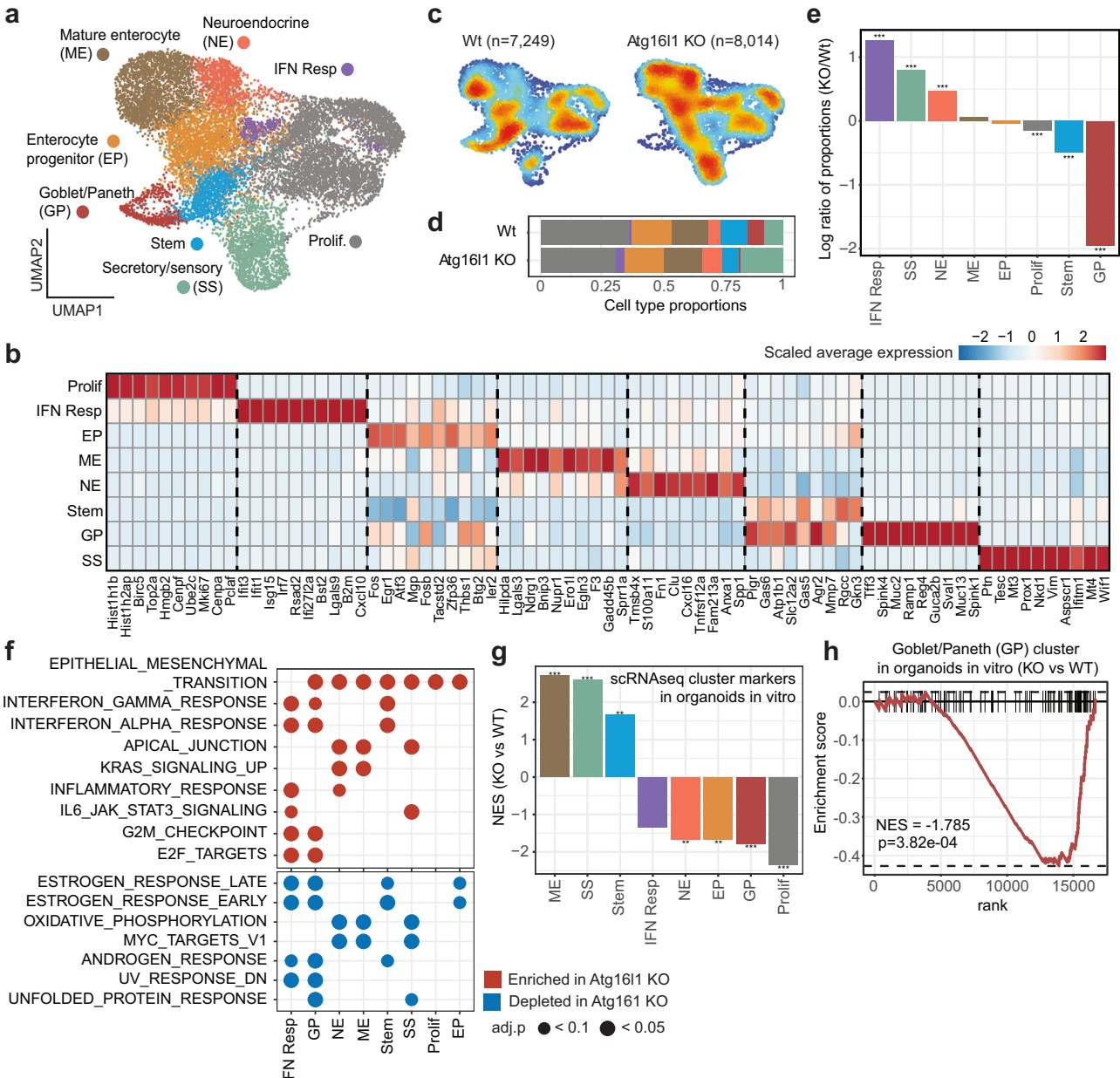

**Fig. 3 | Tumor-intrinsic loss of Atg16l1 alters the phenotype and composition of CRC cells in the liver. a** Visualization of eight scRNA-seq clusters in UMAP dimensions: Prolif (n = 5154), Sensory/secretory (SS) (n = 1956), Mature enterocyte (ME) (n = 2378), Neuroendocrine (NE) (n = 1046), Enterocyte progenitor (EP) (n = 2467), IFN resp (n = 369), Stem (n = 1320), and Goblet/Paneth (GP) (n = 573) clusters. **b**, Heatmap showing the average expression of top 10 markers for each cluster in panel (**a**). **c** Density plot for the WT (left) and Atg16l1 KO (right) conditions. Cells were harvested from n = 2 mice for each condition. Low and high density are denoted by blue and red, respectively. **d** Proportion of cells in each cluster among all cells in the respective sample (Atg16l1 KO or WT). **e** Barplot indicating the proportion change of each cluster in the Atg16l KO sample with respect to the WT condition. For each cluster, the y-axis shows the log-transformed value for (proportion in KO)/(proportion in WT) ratio. Adjusted P-values are derived from two-sided Pearson's chi-squared test for two proportions as

implemented in the prop.test R function, adjusted after false discovery rate correction. **f** GSEA for Atg16l1 KO vs WT contrasts in each cluster using MSigDB Hallmark gene sets. Only gene sets with significant results in at least two contrasts are shown (clusterProfiler, FDR-adjusted P-values for enrichment scores derived via permutation test). **g** GSEA of Atg16l1 KO vs WT organoids in vitro (bulk RNA-seq) for the top 100 genes marking each scRNA-seq cluster. The x-axis denotes clusters while the y-axis shows normalized enrichment scores (normalized to mean enrichment of random samples of the same size). **h** GSEA enrichment plot for GP cluster from panel (**g**). For panels (**g** and **h**), normalized enrichment scores were computed with fgsea using voom+limma derived fold changes; P-values for enrichment scores were derived via permutation test; unadjusted P-values shown. *P < 0.05, **P < 0.01, and ***P < 0.001. (IFN interferon). Source data and exact P-values for panels (**e** and **g**) are provided as a Source Data file.

decrease in the proportion of GP cells in the Atg16l1 KO group, highlighting the vital role of ATG16L1 in maintaining goblet and Paneth cell identity (Fig. 3e). Lastly, Atg16l1 deletion led to a proportional reduction in the Stem[HI] and proliferative (Prolif.) cell clusters (Fig. 3e).

Atg16l1 deletion also altered the transcriptional state of CRC organoid cells within each cluster. Gene set enrichment analysis (GSEA,

Hallmark genesets) revealed that in addition to the dramatically increased IFN responsive cluster, the Stem[HI] and GP cell clusters also exhibited higher levels of IFN response genes in the Atg16l1 KO group, consistent with the generation of an inflammatory tumor microenvironment, increased tumor-intrinsic sensitivity to IFN, or some combination of the two (Fig. 3f). In addition, all clusters except the IFN

responsive cluster showed elevated levels of epithelial-mesenchymal transition (EMT)-associated genes upon Atg16l1 deletion (Fig. 3f). Taking into account the increased proportion of the NE cluster whose identity is strongly associated with EMT-like features, this result highlights EMT as one of the most conspicuous changes in the Atg16l1 KO condition. The above findings suggest that even in immunocompromised NSG hosts, the TME may impart selective pressure on ATG16L1 deficient tumors by enhancing IFN responses.

We next asked which of the transcriptional changes observed in ATG16L1 deficient tumors implanted in vivo were truly tumor-intrinsic. To focus strictly on tumor-intrinsic effects of ATG16L1, we performed bulk RNA-seq profiling on WT and Atg16l1 KO CRC organoids cultured in vitro. GSEA with the top 100 markers of single cell-derived clusters indicated that Atg16l1 deletion led to a significant decrease in the expression of Prolif. and GP cluster markers, and an increase in the expression of SS cluster markers (Fig. 3g, h), demonstrating that these are direct changes caused by the absence of ATG16L1 in tumor cells. In contrast, changes in other clusters (such as Stem^HI, IFN resp and NE) were clearly TME-mediated (Fig. 3e vs. g). Collectively, these data suggest that loss of Atg16l1 leads to a reduction of the goblet, Paneth, stem cell and proliferative pools in CRC organoids, potentially restraining tumor growth. In parallel, enhanced IFN responsiveness may increase anti-tumorigenic immune pressure.

### Loss of Atg16l1 in CRC organoids remodels the tumor myeloid compartment

We next asked how tumor-intrinsic loss of Atg16l1 impacts the hematopoietic tumor microenvironment. Because Atg16l1 KO CRC organoids did not form tumors in immunocompetent BL6 hosts, we limited our analyses to the myeloid compartment of tumors grown in NSG hosts. Here, scRNA-seq analysis was performed on myeloid cells sorted from WT and Atg16l1 KO tumors implanted in the livers of NSG mice, which partitioned into ten clusters ($n = 21,249$) (Fig. 4a, b). Macrophage subsets consisted of TREM2^+, VSIG4^+, MARCO^+, and a mixed cluster of IFN-responsive macrophages/monocytes. Dendritic cells clustered into cDC1, cDC2, CCR7^+ DC, and plasmacytoid DCs. The other two distinct clusters were monocytes and neutrophils. Deletion of Atg16l1 in CRC organoids led to a substantial remodeling of the myeloid compartment. Monocytes as well as TREM2^+ and VSIG4^+ macrophage subsets were significantly reduced in the Atg16l1 KO group (Fig. 4c–e). Neutrophil recruitment was enhanced upon Atg16l1 loss, consistent with increased inflammatory activity even in NSG hosts (Fig. 4e). The majority of macrophage and DC subsets showed evidence of repolarization into an inflammatory state, as suggested by increased IFN response signatures in unbiased GSEA analysis (Fig. 4f). The IFN responsive mixed macrophage/monocyte cluster substantially increased in percentage (Fig. 4e), as well as exhibiting a stronger IFN response in the Atg16l1 KO condition (Fig. 4f). These results indicate that tumor-intrinsic loss of Atg16l1 not only reprograms CRC cells, but also results in pro-inflammatory remodeling of the myeloid compartment towards an anti-tumor state.

### ATG16L1 protects CRC organoids from TNF + IFNγ-mediated programmed cell death

Increased clearance of Atg16l1 KO tumors by cytotoxic lymphocytes and IFNγ (Fig. 2), along with a persistently enhanced IFN-response gene signature in NSG mice (Fig. 3) prompted us to directly characterize the role of tumor-intrinsic ATG16L1 in regulating IFNγ signaling. First, gene expression profiling of WT and Atg16l1 KO CRC organoids was performed following IFNγ stimulation in vitro. Loss of Atg16l1 significantly enhanced IFN pathway gene expression in CRC organoids upon IFNγ treatment (Fig. 5a, b). To test the impact of enhanced IFNγ signaling on tumor cell cytotoxicity, viability of CRC organoids was compared using live cell imaging following treatment with different cytokines

(Supplementary Fig. 10a). In contrast to previous reports showing that autophagy inhibition in highly transformed cell lines or primary intestinal epithelial cells sensitized them to TNF- or IFNγ-induced cytotoxicity[22,46], IFNγ or TNF alone were unable to induce CRC organoid death in vitro (Fig. 5c, d). However, a combination of TNF and IFNγ induced death of CRC organoids. Importantly, loss of Atg16l1 markedly accelerated cell death induced by TNF and IFNγ co-treatment (TNF + IFNγ; Fig. 5c, d and Supplementary Fig. 10b).

We sought to determine whether ATG16L1 also limits cell death initiated by Type I IFN or by engagement of TLR3 and TLR4 (using poly(I:C) and lipopolysaccharide/LPS, respectively). Although Atg16l1 deletion sensitized CRC organoids to cytotoxicity induced by Type I IFN, TLR3, and TLR4, the cell death triggered by a combination of TNF and IFNγ in Atg16l1 KO CRC organoids was more pronounced compared to other combinations (Supplementary Fig. 10c). Finally, we assessed whether other TNF superfamily members, such as TNF-related apoptosis-inducing ligand (TRAIL) and FASL, induced cytoxicity in CRC organoids. No changes in cell death were observed upon stimulation with TRAIL or FASL, either alone or in combination with IFNγ (Supplementary Fig. 10d).

We next investigated the signaling pathways involved in cytokine-mediated cell death. Western blotting revealed the generation of cleaved caspase-3, −8, −11 and Gasdermin-D in both WT and Atg16l1 KO CRC organoids upon TNF + IFNγ co-stimulation (Supplementary Fig. 11a). Intriguingly, the essential necroptosis mediators RIPK3 and mixed lineage kinase domain-like protein (MLKL) were phosphorylated only in Atg16l1 KO organoids (Fig. 5e). Re-expression of ATG16L1 in Atg16l1 KO organoids restored their resistance to TNF and IFNγ-induced cell death (Supplementary Fig. 12a) and attenuated phosphorylation of RIPK3 and MLKL (Supplementary Fig. 12b), confirming that ATG16L1 plays a non-redundant role in limiting the activation of cytokine-induced cell death in these cells. To more thoroughly address the contribution of necroptosis and apoptosis to organoid cell death, Ripk3 was genetically deleted using CRISPR-Cas9 (Supplementary Fig. 11b) and caspases were pharmacologically inhibited with z-VAD in Atg16l1 KO CRC organoids. Deletion of Ripk3 partially rescued TNF + IFNγ-induced death in Atg16l1 KO CRC organoids, and cell death was completely blocked by addition of z-VAD to cells doubly deficient in ATG16L1 and RIPK3 (Fig. 5f). Caspase inhibition in ATG16L1 KO organoids further accelerated cell death, likely via RIPK3 mediated necroptosis, consistent with a regulatory role of caspases in necroptosis[47,48]. Collectively, these results show that absence of Atg16l1 sensitizes CRC organoids to TNF + IFNγ-induced necroptosis and apoptosis.

## Discussion

CRC is one of the most commonly diagnosed cancers and a leading cause of cancer-associated mortality worldwide[49–51]. Despite the success of checkpoint blockade in solid tumors, non-MSI CRC has consistently failed to demonstrate an appreciable response to immunotherapy. In this study, we leveraged GWAS of inflammatory bowel disease to focus on ATG16L1, a gene highly associated with intestinal inflammation in Crohn's disease. Analysis of core members of the autophagosome elongation machinery (ATG3, ATG4A-D, ATG5, ATG7, ATG10, ATG12, ATG16L1) revealed a specific association of decreased ATG16L1 gene expression with improved immunotherapy-dependent outcome of non-MSI CRC harboring KRAS mutations. Consistently, ATG16L1 exhibited a clear enrichment in CRC epithelium when compared with the above ATG genes. Nevertheless, detectable expression of ATG16L1 in non-epithelial cells (particularly T cells) from a minority of samples could confound the interpretation of bulk tissue transcriptome data, and future histopathological studies of ATG16L1 protein expression in large patient cohorts should further clarify the association of epithelial ATG16L1 expression with immunotherapy outcome.

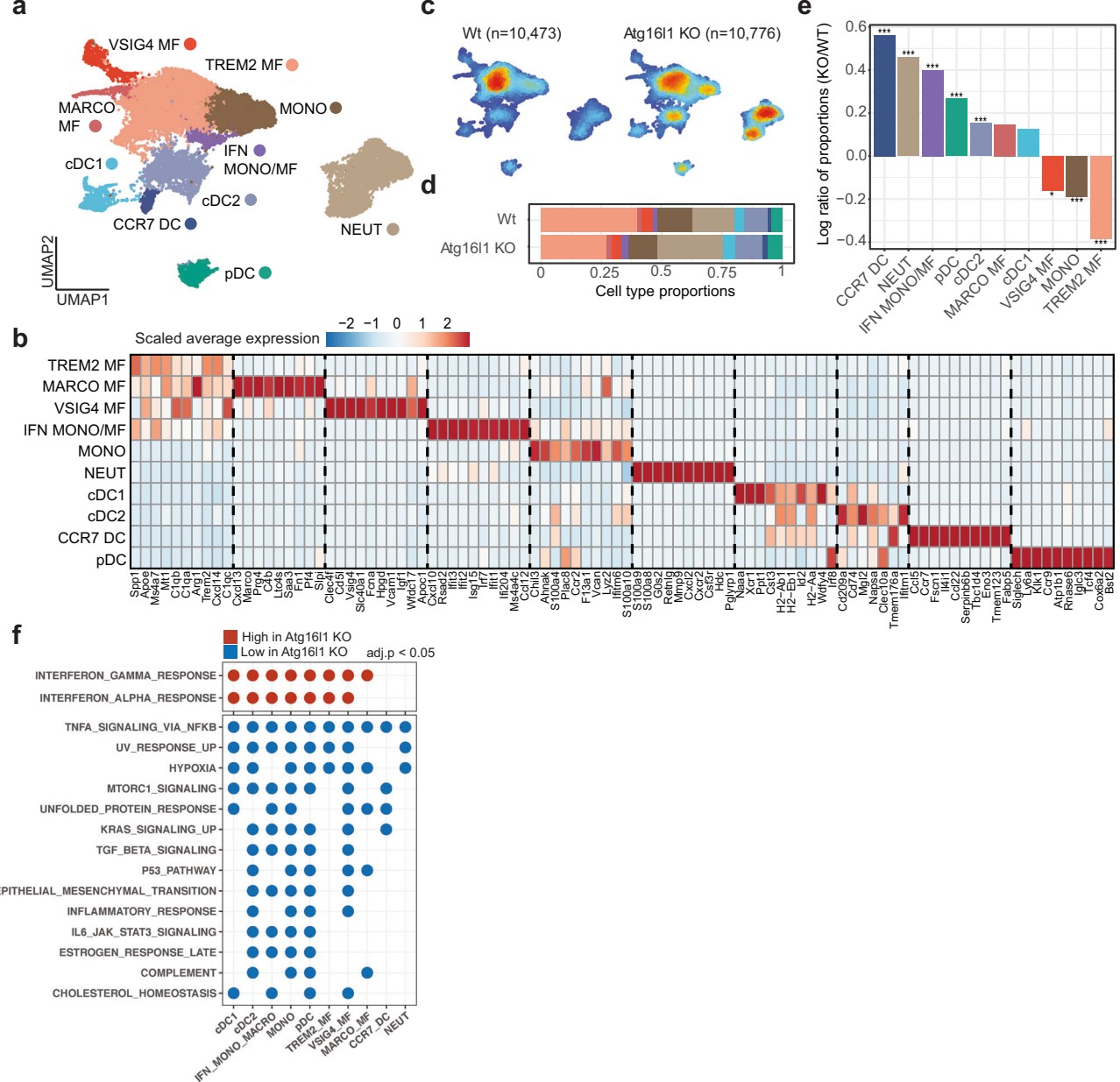

**Fig. 4 | Tumor-intrinsic loss of Atg16l1 reprograms myeloid cell composition and phenotypes in the tumor microenvironment. a** Visualization of ten scRNA-seq clusters in UMAP dimensions: TREM2 MF ($n = 7116$), VSIG4 MF ($n = 890$), MARCO MF ($n = 434$), IFN responsive MONO/MF ($n = 560$), MONO ($n = 2767$), cDC1 ($n = 969$), cDC2 ($n = 2297$), CCR7⁺ DC ($n = 331$), pDC ($n = 1115$), NEUT ($n = 4770$) clusters. **b** Heatmap showing the average expression of top 10 markers for each cluster in panel (**a**). **c** Density plot for the WT (left) and Atg16l1 KO (right) groups. Cells were harvested from $n = 2$ mice for each condition. Low and high density are denoted by blue and red, respectively. **d** Proportion of cells in each cluster among all cells in the respective sample (Atg16l1 KO or WT). **e** Barplot indicating the proportion change of each cluster in the Atg16l1 KO sample compared to the WT

group. For each cluster, the *y*-axis shows the log-transformed value for (proportion in KO)/ (proportion in WT) ratio. *P*-values were calculated from two-sided Pearson's chi-squared test for two proportions as implemented in the prop.test R function, and adjusted by false discovery rate correction. **f** GSEA for Atg16l1 KO vs WT contrasts in each cluster using MSigDB Hallmark gene sets. Only gene sets with significant results in at least four contrasts are shown (clusterProfiler, *P*-values for enrichment scores derived via permutation test, FDR-adjusted *P*-values < 0.05). \**P* < 0.05, \*\**P* < 0.01, and \*\*\**P* < 0.001. (MF macrophages, MONO monocytes, DC dendritic cells, pDC plasmacytoid dendritic cells, NEUT neutrophils). Source data and exact *P*-values for panel **e** are provided as a Source Data file.

Using a model of MSS-CRC, we demonstrated that loss of Atg16l1 drove productive cellular immunity against MSS-CRC tumors, enhanced IFN signaling, and accelerated programmed cell death. ATG16L1 emerged as a critical cytoprotective factor that suppressed IFN responses both in vivo and ex vivo; this is likely a key – but not sole – mechanism by which autophagy and ATG16L1 promote CRC fitness, as IFN signaling is known to license multiple cell death programs (reviewed in[52,53]). We and others have previously shown that ATG16L1 deficiency in myeloid cells or the intestinal epithelium enhanced

programmed cell death in an IFN-dependent manner[24,52,54]. While ATG16L1 deficient myeloid cells demonstrated accumulation of innate inflammatory factors such TRIF and ZBP1[24], this was not observed in ATG16L1 deficient CRC organoids (data not shown). Phosphorylation of necroptosis regulators RIPK3 and MLKL and cleavage of apoptotic caspases were enhanced upon loss of Atg16l1. The molecular mechanism(s) by which autophagy or ATG16L1 suppress IFN-dependent licensing of cell death in intestinal epithelial cells remains poorly understood, requiring focused investigation.

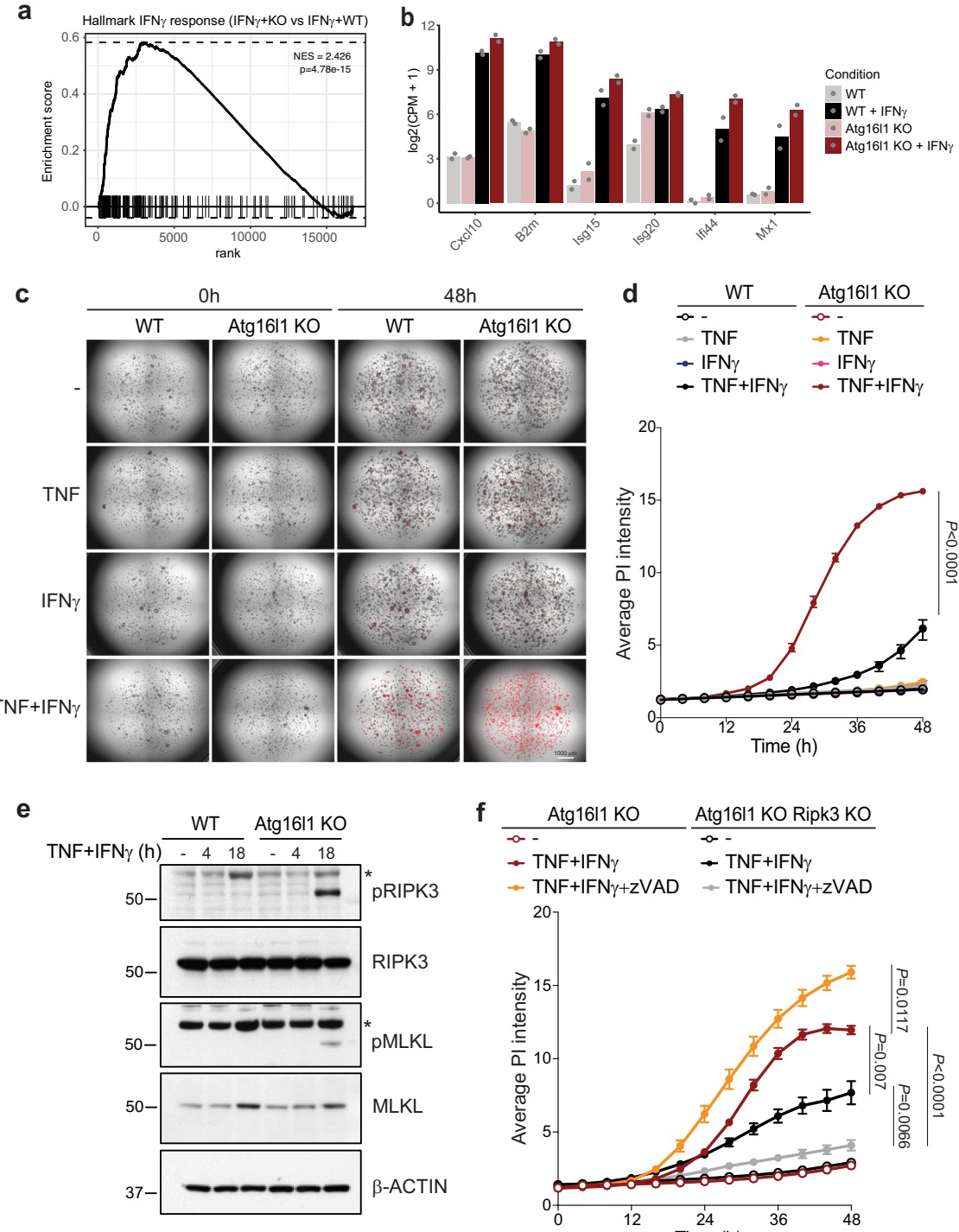

**Fig. 5 | Atg16l1 loss sensitizes CRC organoids to TNF + IFNγ-induced cell death.**
**a** GSEA using MSigDB IFNγ response hallmark gene set for differential expression between Atg16l1 KO (n = 2) and WT (n = 2) CRC organoids treated (in vitro bulk RNA-seq data) with p-value = 4.78 × 10$^{-15}$ (normalized enrichment score was computed with fgsea using voom+limma derived fold changes; P-value for enrichment score was derived via permutation test). **b** Differential expression of representative IFNγ-induced genes between Atg16l1 KO (n = 2) and WT (n = 2) CRC organoids in either untreated or IFNγ-treated conditions. **c** WT or Atg16l1 KO CRC organoids treated as indicated and stained with propidium iodide (PI). Scale bar = 1000 μm. Images are representative of 4–6 technical replicates per condition (see Source

Data for Fig. 5d). **d**, **f** Cell death assayed by live imaging of WT versus Atg16l1 KO organoids (**d**), and Atg16l1 KO versus Atg16l1 + RIPK3 double KO organoids (**f**) treated with combinations of TNF + IFNγ + zVAD for 48 h. PI staining is measured by fluorescence intensity/μm². Groups compared using two-way ANOVA.
**e** Immunoblot analysis of the indicated phosphorylated and total proteins in WT or Atg16l1 KO CRC organoids stimulated with combinations of TNF + IFNγ for 4 or 18 h. * indicates non-specific bands. Data for panels **c**–**f** are representative of three independent experiments. Summary data are shown as mean ± s.e.m. Source data for panels (**d**–**f**) are provided as a Source Data file.

IMblaze370 is a unique Phase III clinical trial investigating the potential of checkpoint blockade in advanced metastatic CRC, particularly the highly immunotherapy-resistant microsatellite-stable subset. In this well-powered study, we find that *ATG16L1* expression is a predictive biomarker of poor response to the PD-L1 inhibitor atezolizumab, specifically in patients harboring oncogenic KRAS mutations. While the mechanistic basis of this finding remains to be determined, it is consistent with previously observed dependency of KRAS-mutant tumors on autophagy as a mechanism of therapeutic resistance and in vivo fitness[55–61]. A lack of independent clinical datasets to validate these encouraging observations presents a limitation to the study. We acknowledge that the data must be considered as an initial finding, warranting confirmation in follow-up studies with comparable patient populations and treatment arms.

Accumulating evidence of the immuno-modulatory and pro-tumorigenic roles of autophagy has renewed interest in pharmacological inhibition of this pathway to reverse therapeutic resistance in cancer (reviewed in[62]). For example, RAS pathway inhibition in pancreatic ductal adenocarcinoma (PDAC) revealed a dependence on autophagy that could be exploited therapeutically, prompting clinical investigations of lysosomal inhibitors such as hydroxychloroquine (HCQ) or chloroquine (CQ) in combination with standard of care chemotherapies[55,56]. Whether these combinations impact anti-tumor immunity in clinical settings remains to be determined, but assessment of murine models suggests that short-term treatment with CQ did not compromise host immunity against immunogenic tumors[63]. Modulation of antigen presentation was also proposed as a tumor-intrinsic mechanism by which autophagy promotes immunosuppression in models of PDAC, although this remains to be recapitulated in other settings[26]. Intriguingly, we observed that NK cells and host-derived IFN-γ, but not CD8+ T cells, were primary drivers of cytotoxicity against ATG16L1-deficient tumors. This posits an antigen-independent mechanism of anti-tumor immunity in our model, and is consistent with a lack of tumor mutation burden (TMB) in MSS-CRC, a basis for generally poor immunotherapy responses in CRC[64,65]. Our findings thus expand the scope of autophagy modulation as a therapeutic avenue in MSS-CRC.

Beyond tumor-intrinsic autophagy, pre-clinical models have demonstrated immunostimulatory effects of systemic autophagy inhibition either using genetic or pharmacological models. For instance, small molecule inhibitors for the class III PI3 kinase VPS34, an upstream regulator of autophagy, or lysosomal inhibition with CQ have demonstrated increased T and NK cell infiltration in tumors along with effector cytokine release and improved antigen presentation via MHC-I[26,30]. Several actionable targets within the autophagic flux program make it an attractive pathway for therapeutic modulation; however, caution needs to be exercised when considering regulators of membrane trafficking (e.g., VPS34, Unc51-like kinases ULK1/2) or lysosomal fitness (e.g., CQ/HCQ, other lysosomal inhibitors). These represent master regulatory nodes that impact multiple processes beyond autophagy, and inhibitors of these targets may thus be limited by toxicity. Germline knockout models provide further evidence of pathway divergence. While loss of core members of the autophagosome elongation machinery Atg3, 5, 7, 10, 12 and Atg16l1 demonstrate neonatal lethality, deletion of Beclin 1, Vps34, Rb1cc1 result in embryonic lethality. Single deletion of Ulk1 or Ulk2 appears to be tolerated, but combined deletion results in neonatal lethality associated with both autophagy-dependent and independent processes[66–69]. Sustained pharmacological inhibition of VPS34 is also shown to be poorly tolerated, consistent with its essential role in mammalian development[70]. Finally, while multiple clinical trials are currently investigating CQ/HCQ in cancer, there lacks mechanistic demonstration that any therapeutic benefit conferred by these lysosomal inhibitors is primarily due to autophagy inhibition. These findings highlight the need to focus on more specific modulators of

autophagosome formation, for example members of the ATG16L1 complex and the de-ubiquitinases of the ATG4 family which recycle LC3 to sustain autophagic flux.

Our current study indicates that high *ATG16L1* expression predicts poor immunotherapy response in CRC, which can be explained mechanistically by ATG16L1-mediated inhibition of IFN signaling and a consequent suppression of cellular immunity. This provides a therapeutic rationale for autophagy inhibition to overcome immunotherapy resistance in advanced MSS-CRC, a disease with limited treatment options, poor clinical outcomes and high unmet need[18,71,72].

# Methods

## Mice
WT B57BL/6 mice (000664), NOD.Cg-Prkdcscidll2rgtm1Wjl/SzJ (NSG) (colony 005557) and B6.129S7-IFNgtm1TS/J (*Ifng KO*) (002287) mice were purchased from the Jackson Laboratory. Females of 6 to 12 weeks old were used for experiments. All mice were housed at Genentech in individually ventilated cages within animal rooms maintained on a 14:10-h, light:dark cycle. Animal rooms were temperature and humidity-controlled, between 68 and 79 °F and 30 and 70%, respectively, with 10 to 15 room air exchanges per hour. Mice were acclimated to study conditions for at least 3 days before tumor cell implantation. Animal studies were approved by Genentech's Institutional Animal Care and Use Committee and adhere to the NRC Guidelines for the Care and Use of Laboratory Animals. In all studies, mice were monitored daily for adverse clinical signs, and animals were euthanized for human reasons if they exhibited body weight reduction >15% or developed rectal prolapse.

## Hydrodynamic tail-vein (HTV) injection of tumor cells
Animals were restrained without anesthesia in a conical acrylic restrainer with a heating element to dilate blood vessels. Each mouse was injected intravenously in the tail vein with 1.8 mL of the solution containing 50,000 cells in PBS in a single dose administered as a bolus intravenous injection (tail vein) over 4 to 5 s (8 s maximum). Animals were observed continually for adverse clinical signs for at least 15 min after dose.

## Intravenous tail vein (IV) injection of tumor cells
Animals were restrained without anesthesia in a conical acrylic restrainer with a heating element to dilate blood vessels. Each mouse was injected intravenously in the tail vein with 100 μL of the solution containing 50,000 cells in Hank's balanced salt solution (HBSS).

## Orthotopic injection of tumor cells
The lumen implantation procedure has previously been described[36]. Briefly, mice were anesthetized by isoflurane inhalation and injected subcutaneously with buprenorphine at 0.05 to 0.1 mg/kg. A blunt-ended hemostat (Micro-Mosquito, No. 13010-12, Fine Science Tools) was inserted ~1 cm into the anus. The hemostat was angled toward the mucosa and opened slightly such that a single mucosal fold could be clasped by closing the hemostat to the first notch. The hemostat was retracted from the anus, exposing the clasped exteriorized mucosa. 10 μL of solution containing 50,000 cells admixed with 50% matrigel (Corning) in PBS was injected directly into the colonic mucosa. The hemostat was released after reversing the prolapse. At the study endpoint, colons were resected and tumor dimensions were measured using calipers. Tumor volume was calculated with the following formula: tumor volume (mm$^3$) = (length × width$^2$) × 0.5.

## Bioluminescence imaging
Mice were anesthetized using isoflurane, injected i.p. with 200 μL of 25 mg/mL D-luciferin (Goldbio, LUCNA-100) and imaged on the Lago Imaging System (Spectral Instruments Imaging). During image

acquisition, animals received anesthesia from a nose cone delivery system while their body temperatures were maintained on a thermostatically controlled platform. Photon counts per second of the observational area were calculated using Aura software (Spectral Instruments).

## In vivo depletion of CD8+ T cells and NK cells

To deplete CD8+ T and NK cells, mice were treated with indicated antibodies two times a week throughout the duration of the study. Depletion was initiated one day before tumor inoculation by I.P. administration of 10 mg/kg anti-CD8 IgG2b (clone 2.43; produced at Genentech) and 5 mg/kg anti-mouse NK1.1 IgG2a (clone PK136; cat # BE0036, Bio X Cell, Lebanon, NH) or isotype control mouse IgG2b (Genentech) and mouse IgG2a (Genentech). To monitor depletion efficiency, non-competing antibodies were used for flow cytometry analysis, as indicated in the flow cytometry section.

## Western Blotting

CRC organoids pellets were lysed in RIPA buffer and protease and phosphatase inhibitors (Roche) at 4 °C for 20 min. Supernatants were obtained after high-speed centrifugation and protein concentration measured using the BCA assay (Thermo Fisher). Lysates were denatured with reducing sample buffer and dithiothreitol (Invitrogen) at 95 °C for 10 min. Proteins were separated by sodium dodecyl sulfate-polyacrylamide gel electrophoresis (NuPAGE) (4–12% gradient Bis-Tris gel) and analyzed by western blotting with antibodies against ATG16L1 (1:4000; clone 1F12; cat # M150-3, MBL International, Woburn, MA), MLKL (1:1000; clone 1G12; Genentech), pSer345 MLKL (1:1000; clone D6E3G; cat # 37333, Cell Signaling Technology, Danvers, MA), RIPK3 (1:2000; cat # NBP1-77299, Novus Biologicals, Centennial, CO), pThr231/Ser232 RIPK3 (1:1000; clone GEN-135-35-9; Genentech), GSDMD (1:4000; clone GN20-13; Genentech), CAL-COCO1 (1:4000; cat # 19843-1-AP, Proteintech, Rosemont, IL), Sqstm1/p62 (1:4000; cat # 5114, Cell Signaling Technology, Danvers, MA), TAX1BP1 (1:4000; clone EPR13287(B); cat # ab176572, Abcam, Waltham, MA), LC3I/II (1:4000; clone D3U4C; cat # 12741, Cell Signaling Technology, Danvers, MA), cleaved Caspase-3 (1:2000; clone 5A1E; cat # 9664, Cell Signaling Technology, Danvers, MA), Caspase-8 (1:1000; clone D35G2; cat # 4790, Cell Signaling Technology, Danvers, MA), Caspase-11 (1:1000; clone 17D9; cat # 14340, Cell Signaling Technology, Danvers, MA), β-actin (1:10000; clone D6A8; cat # 8475, Cell Signaling Technology Technology, Danvers, MA), rabbit IgG-HRP (1:4000; cat # 7074, Cell Signaling Technology, Danvers, MA), mouse IgG-HRP (1:4000; cat # 7076, Cell Signaling Technology, Danvers, MA), rat IgG-HRP (1:4000; cat # 7077, Cell Signaling Technology, Danvers, MA). Validation data for commercially available antibodies can be found on vendor websites. Antibodies generated at Genentech have been validated previously using knockout cell lines[24].

## Tissue processing

Spleens were minced on a 70 μm nylon filter (Corning). The flow-through was collected and centrifuged at 1500 rpm at 2–8 °C. The supernatant was then aspirated and the cell pellets incubated with 5 mL (spleen) of ACK Lysis Buffer for 5 min at room temperature and followed by two washes with cold PBS. Cell pellets were then resuspended in a staining buffer. To generate single-cell suspensions of tumor tissue, liver tumor nodules were dissected and minced into 2- to 4-mm pieces. Next, tissues were placed in a 15 ml conical tube with 3 ml digestion medium (RPMI + 2% FBS with 100 mg/ml Dispase (Life Tech), 100–200 mg/ml collagenase P (Roche), and 50 mg/ml DNase I (Roche) and agitated. Tubes were placed in a 37 °C water bath for 15 min, and solutions were removed and filtered (70 μm) into RPMI supplemented with 2% FCS (VWR). The digestion was repeated three times. Tumor single-cell suspension was then resuspended in a staining buffer.

## Flow cytometry

Single-cell suspensions were incubated in anti-CD16/CD32 Fc block antibody (5 μg/ml; clone 2.4G2; cat # 553141, BD Biosciences, San Jose, CA) and fixable viability dye eFluor 780 (Invitrogen, Carlsbad, CA) for 15 min at 4 °C in cold PBS. Cells were then washed once and stained with combinations of the following antibodies: CD45-BV510 (2 μg/ml; clone 30-F11; cat # 103138, BioLegend, San Diego, CA), TCRβ-PerCP-Cy5.5 (2 μg/ml; clone H57-597; cat # 109228, BioLegend, San Diego, CA), CD4-BUV395 (2 μg/ml; clone GK1.5; cat # 563790, BD Biosciences, San Jose, CA), CD8-af700 (5 μg/ml; clone KT15; cat # MCA609A700, Bio-Rad, Hercules, CA), NKp46-APC (1 μg/ml; clone 29A1.4; cat # 137607, BioLegend, San Diego, CA), CD49b-PE (2 μg/ml; clone DX5; cat # 108908, BioLegend, San Diego, CA), B220-BUV661 (2 μg/ml; clone RA3-6B2; cat # 612972, BD Biosciences, San Jose, CA), and Thy1.2-BUV805 (1 μg/ml; clone 53-2.1; cat # 741908, BD Biosciences, San Jose, CA). Data were acquired on a Symphony flow cytometer (BD Biosciences, San Jose, CA) and analyzed using FlowJo software (FlowJo LLC, Ashland, OR). Gating strategies for the identification of CD8+ T cells and NK cells are provided in Supplementary Fig. 13.

## Organoid growth assay via Incucyte

CRC organoids were dissociated in a cell dissociation buffer (Gibco) for 10 min at 4 °C, washed in PBS, centrifuged and treated with Accutase (Sigma-Aldrich) for 5 min at 37 °C. Next, cells were washed with PBS and filtered through 70 μM cell strainer. Cells were counted and approximately 7000 cells were seeded into a 48 well plate in 50% Matrigel and 50% culturing media containing advanced DMEM/F12 medium (Invitrogen), B27 (Thermo fisher), N2 (Thermo fisher) and N-acetylcysteine (Sigma-Aldrich) to form a dome. The plate was placed in a 37 °C incubator for at least 20 min and then 250 μL of culturing media per well was added. The plate was placed into the Incucyte® SX5 Live-Cell Analysis System (Essen Bioscience) and was imaged and analyzed using the Organoid Culture QC module. In brief, whole well brightfield images were acquired with a 4x objective every four hours for a total of 72 h. Organoids were masked and the total area was measured at each timepoint using the Incucyte organoid analysis software.

## Generation of AKPS CRC organoids

Colon from adult BL6 mice was removed, flushed, opened lengthwise and washed in cold PBS to remove all luminal contents. The colon was cut into 0.5-1 cm pieces in cold PBS, vortexed, washed for 3 times and placed into 25 ml 2.5 mM EDTA-PBS for 5 min at 37 °C. The supernatant was then removed and the colon pieces were washed with PBS followed by the incubation in 25 ml 5 mM EDTA-PBS for 15 min at 37 °C. After being vigorously vortexed, supernatant was collected and was filtered through 100 μm filters, centrifuged at 500 g for 5 min, washed with cold PBS and then resuspended with 1:1 in Intesticult media (StemCell) and Matrigel (Corning). Intesticult media was changed on organoids every 2-3 days and they were passaged by mechanical disruption every 7–10 days.

To generate AKPS organoid mutations in Apc, KrasG12D, Trp53 and Smad4 were introduced by CRISPR-Cas9 technology. The following gene-specific sgRNAs were used: Apc, CAGGACTGCATTCTCCT-GAA, AATGCAGTCCTGTCCCCATG, TTCTTGGGAATGACCCCATG; Kras, CTGAATTAGCTGTATCGTCA and G12D donor sequence 5′-ATG TTC TAA TTT AGT TGT ATT TTA TTA TTT TTA TTG TAA GGC CTG CTG AAA ATG ACT GAG TAT AAA CTT GTG GTG GTT GGA GCT GAT GGC GTA GGC AAG AGC GCA TTG ACG ATA CAG CTA ATT CAG AAT CAC TTT GTG GAT GAG TAT GAC CCT ACG ATA GAG GTA ACG CTG CTC TAC AGT CTG CGT GCG C-3′; Trp53, GGAGCTCCTGACACTCGGAG; and Smad4, GATGTGTCATAGACAAGGTG. Organoids were dissociated into a single-cell suspension using accutase (Sigma-Aldrich) for 5 min at 37 °C and then electroporated with 2 μL of Cas9 and 3 μl of sgRNA using the P1 buffer and CM137 program (Lonza). After electroporation

cells were embedded in Matrigel and DMEM advanced (GIBCO) media supplemented with 10 mM Hepes (Sigma-Aldrich), 2 mM GlutaMAX (Life Technologies), 1x Penicillin/Streptomycin (Life Technologies), 1x N2 (GIBCO), 1x B27 (GIBCO), 1 mM N-acetysteine (Sigma Aldrich), 50 ng/ml EGF (Life Technologies), 100 ng/ml Noggin (Peprotech), and R-spondin-1 (R&D systems). Selection for mutated cells was performed using growth factor depletion from the culture medium (R-spondin depletion to select for Apc deficient cells, EGF depletion to select for Kras + /G12D cells, presence of 10 μM Nutlin-3 (Sigma-Aldrich) to select for Trp53 mutants, and Noggin depletion with presence of TGF-β (Peprotech, 10 ng/ml) to select for cells with Smad4 deficiency. AKPS were also electroporated under the same conditions described above to express GFP and luciferase with a pB1-EF1-Luciferase-T2A-GFP vector (PiggyBac Transposon, System Biosciences). GFP + AKPS cells were flow sorted for downstream use.

### Generation of Atg16l1 KO and Ripk3 KO CRC organoids
To delete Atg16l1, AKPS CRC organoids were electroporated with the sgRNA sequences ACTGCACAAGAAGCGTGGGG and GGGTCTGGTT GGCTACCTCG using same conditions described in the paragraph above. To generate Ripk3 KO the sgRNA sequences GCCCGGACA CGAAGTCCCAC and GCGGAGGGTTCAAGCTGTGT were used. Three days after electroporation, CRC organoids were dissociated in cell dissociation buffer (Gibco) for 10 min at 4 °C, washed in PBS, centrifuged and treated with accutase (Sigma-Aldrich) for 5 min at 37 °C. Next, cells were washed with PBS and filtered through 70 μM cell strainer. Viable single-cell suspensions were sorted into 96 well plate flat bottom pre-coated with 50% Matrigel (Corning) to generate single cell clones. Clones producing colonies were tested for ATG16L1 expression by Western blotting.

### Re-expression of Atg16l1 in Atg16l1 KO CRC organoids
Atg16l1 re-expression plasmid (pb-EF1_Atg16l1OE-IRES_tagBFP2) was generated with mouse Atg16l1 sequence (isoform 1, https://www.uniprot.org/uniprot/Q8C0J2) into pb1-EF1-IRES-tagBFP2 piggybac vector. pb-EF1_Atg16l1OE-IRES_tagBFP2 (2 μg) or its control vector pb1-EF1_tagBFP2 (2 μg) together with transposase plasmid (1 μg) was electroporated into AKPS-GFP-Luc-ATG16.ko organoids using the P1 buffer and CM137 program (Lonza). BFP+ cells were flow sorted to generate a stably expressed Atg16l1 cell line for the downstream use.

### Cell death assay
CRC organoids were dissociated in a cell dissociation buffer (Gibco) for 10 min at 4 °C, washed in PBS, centrifuged and treated with Accutase (Sigma-Aldrich) for 5 min at 37 °C. Next, cells were washed with PBS and filtered through 70 μM cell strainer. Cells were counted and approximately 7000 cells were seeded into a 48 well plate in 50% Matrigel and 50% culturing media containing advanced DMEM/F12 medium (Invitrogen), B27 (Thermo fisher), N2 (Thermo fisher) and N-acetylcysteine (Sigma-Aldrich) to form a dome. The plate was placed in a 37 °C incubator for at least 20 min and then 250 μl of culturing media per well was added. After two days, medium was replaced with medium containing 100 ng/ml recombinant murine IFNγ (R&D Systems), 50 ng/ml mTNF (Peprotech),1 μg/ml LPS (Invivogen), 2 μg/ml Poly(I:C) (Invivogen), 100U/ml IFNa (PBL), 100U/ml IFNb (PBL), 50 ng/ml TRAIL (produced in house), 50 ng/ml CD95L (produced in house) 50 μM Z-VAD-FMK (Promega), 2 μg/ml Propidium Iodide (PI, Invitrogen). CRC organoids were imaged every 4 h for a total of 48 h with a 4x Plan Fluor objective (NA: 0.13, Nikon) on a Nikon Ti-E inverted microscope equipped with a Neo scMOS camera (Andor, Oxford Instruments), a linear encoded automated stage (Applied Scientific Instrumentation), 37 °C/5% CO₂ environmental chamber (Okolab), all run by NIS Elements software (Nikon). Organoids were imaged in TRITC and Brightfield channels. Images covering the whole matrigel

plug (2 × 2 field of views, 3 × 200 μm Z steps) were stitched and focused into one image projection with an extended depth of focus module (EDF; Nikon). A custom script (Matlab, Mathworks) was used for cell death measurements. Organoids were identified and masked in brightfield image. The organoid mask was then applied to the TRITC image, fluorescence intensity (FI) of the propidium iodide was measured within the mask and was normalized to total organoid area (FI/μm²).

### Histology and Image Analysis
Whole organs were fixed in 10% neutral buffered formalin for 48 h prior to being processed and embedded into paraffin blocks utilizing standard protocols. 4 micron paraffin sections were cut onto charged glass slides and stained with hematoxylin and eosin (H&E) utilizing an autostainer following standard protocols. H&E stained slides were scanned on a NanoZoomer X360 whole slide imager (Hamamatsu, Bridgewater NJ) at 200x final magnification. Tumor nodules were manually annotated using a web-based image viewer platform. Automated image analysis algorithms were applied in Matlab (r2019a, Mathworks, Natick, MA) to determine tumor nodule area and total tissue area.

### Immunohistochemical staining of ATG16L1
ATG16L1 protein was detected by immunohistochemical staining of formalin-fixed paraffin-embedded tissue per standard Discovery XT Autostainer protocols. In brief, slides were deparaffinized and subjected to antigen retrieval (Ventana CC1). Primary antibody was applied (mouse anti-ATG16L1 antibody from MBL, clone 1F12, catalog# M150-3) followed by secondary antibody (Ventana Mouse OmniMap) and DAB chromogen. Samples were scored by a pathologist (manual visual assessment, AS) in the following manner: intensity of tumor epithelial staining was scored from 0–4, with an associated prevalence score (by area), to yield an overall H-score as follows: (H1*area%) +(H2*area)+(H3*area). Qualitative staining of other tissue compartments (non-neoplastic epithelium, and intra-tumoral stromal fibroblasts and vascular endothelium) was also noted. Samples were excluded from the analysis if staining quality was poor (technical staining failure) or if insufficient tumor epithelium was present on the slide.

### Statistical analysis
GraphPad Prism seven was used for data analysis and representation. Pairwise statistical analyses with appropriate multiple testing corrections were performed as described in figure legends. All tests were two-sided with a significance threshold of 0.05. Line graphs and associated data points represent means of data. Data shown in graphs represent mean values ± s.e.m. Box and whisker plots are defined in figure legends.

### Whole-exome sequencing (WES)
Exome capture was performed using Agilent's SureSelect Mouse 50 Mb baits and processed using the SureSelect version 1.5.1 protocol. Whole-exome sequencing (WES) was performed on Illumina HiSeq 2500 sequencers to obtain 75 bp paired-end reads with an average of 32 million and 111 million fragments per sample, respectively. Reads were aligned to the mouse reference genome (NCBI Build 38) using GSNAP37 version 2013-10-10, allowing a maximum of two mismatches per 75-base sequence (parameters: -M 2 -n 10 -B 2 -i 1−pairmax-dna = 1000−terminal-threshold = 1000−gmap-mode = none−clip-overlap).

### TCGA analysis
The results shown here are in whole or part based upon data generated by the TCGA Research Network: http://cancergenome.nih.gov/.

Normalized and batch-adjusted RNA-seq data were obtained from the PanCanAtlas publications page of the Genomic Data Commons (https://gdc.cancer.gov/about-data/publications/pancanatlas).

### Computation of signature scores

In bulk RNA-seq, signature scores were computed with the mean z-score approach. In this approach, values for each gene are first z-transformed across all samples. Resulting z-scores are then averaged across genes to arrive at a single signature score for each sample. In single cell RNA-seq, signature scores were computed with the AddModuleScore function in Seurat (v 4.0.2).

### Survival Analysis

Kaplan Meier plots and log-rank *P*-values were generated with the survminer package in R. This package uses the survdiff function for log-rank P-values. We obtained hazard ratios and 95% confidence intervals using the coxph R function.

### Bulk RNA sequencing of in vitro cultured CRC organoids

Around 500,000 AKPS organoid cells were collected per condition and RNA was isolated according to the manufacturer's protocol (Qiagen, 74104). Total RNA was quantified with Qubit RNA HS Assay Kit (Thermo Fisher Scientific) and quality was assessed using RNA ScreenTape on 4200 TapeStation (Agilent Technologies). For sequencing library generation, the Truseq Stranded mRNA kit (Illumina) was used with an input of 100 nanograms of total RNA. Libraries were quantified with Qubit dsDNA HS Assay Kit (Thermo Fisher Scientific) and the average library size was determined using D1000 ScreenTape on 4200 TapeStation (Agilent Technologies). Libraries were pooled and sequenced on NovaSeq 6000 (Illumina) to generate 30 million single-end 50-base pair reads for each sample. RNA-sequencing data were analyzed using HTSeqGenie[73] (https://bioconductor.org/packages/HTSeqGenie/) in BioConductor[74] as follows: first, reads with low nucleotide qualities (70% of bases with quality <23) or rRNA and adapter contamination were removed. The reads that passed were then aligned to the mouse reference genome GRCm38.p5 using GSNAP[75] version '2013-10-10-v2' allowing maximum of two mismatches per 75 base sequence (parameters: '-M 2 -n 10 -B 2 -i 1 -N 1 -w 200000 -E 1 --pairmax-rna=200000'). Transcript annotation was based on the GENCODE M15. To quantify gene expression levels, the number of reads mapping unambiguously to the exons of each gene was calculated. Gene expression levels were quantified as Reads Per Kilobase of exon model per Million mapped reads normalized by size factor (nRPKM), defined as number of reads aligning to a gene in a sample / (total number of uniquely mapped reads for that sample x gene length x size factor). Differential expression and gene set enrichment analysis were performed with voom + limma[76] and fgsea[77] respectively.

### Single cell RNA sequencing of CRC organoids implanted in vivo

**Library prep**. AKPS-eGFP⁺Luc⁺ CRC organoids were modified (CRISPR-Cas9) to delete ATG16L1. Atg16l1 KO and WT cells were inoculated into the liver of NSG mice via HTV injection. Tumors were harvested following ~3 weeks of growth and live Calcein blue⁺ 7ADD⁻ cells were sorted into tumor (eGFP⁺CD45⁻) and immune (eGFP⁻CD45⁺) cells. A total of 4 samples (Atg16l1 KO and WT for each sort) were submitted for processing. Single-cell suspensions were converted to scRNA-seq libraries using the 10x Genomics Chromium Single Cell 3′ Library, Gel Bead & Multiplex Kit (v3.1 chemistry) following manufacturer's instructions. The final libraries were profiled using the Bioanalyzer High Sensitivity DNA Kit (Agilent Technologies) and quantified using the Kapa Library Quantification Kit (Kapa Biosystems). 138 base length paired-end sequencing was performed for each single-cell RNA-seq library in one lane of NovaSeq 6000 (Illumina).

**Data preprocessing**. Demultiplexing of raw data was performed using 10X Genomics CellRanger mkfastq software. Reads obtained from demultiplexing were used as the input for 'cellranger count' (CellRanger v6.0.1), which aligned the reads to the mouse reference genome GRCm38.p5 using STAR and collapsed to unique molecular identifier (UMI) counts. Resulting gene-barcode matrices were filtered for background noise with 10X's cell-calling heuristic. The filtered gene-barcode matrices only included cells with at least 500 UMI counts. Subsequent data analysis was carried out in R 4.1.0 and the Seurat package (v4.0.4). Before quality control, the number of tumor and immune cells in the dataset were 21657 and 23322, respectively.

Quality control steps for tumor cells included: 1) Low quality cells were removed based on high mitochondrial content (>7.5%) or low number of detected genes (<500). 2) Potential doublets were removed based on high number of detected genes (>7200). 3) Erythrocytes were removed based on *Hbb-bs*, *Hba-a1*, *Hbb-bt* expression. 4) Endothelial cells were removed based on *Pecam1*, *Esam*, *Cd93*, *Cdh5* expression. Number of tumor cells after quality control was 15263 (7249 and 8014 from the WT and Atg16l1 KO sample respectively).

Quality control steps for immune (myeloid) cells included: (1) Low quality cells were removed based on high mitochondrial content (>5%) or low number of detected genes (<200 to keep neutrophils). (2) Potential doublets were removed based on high number of detected genes (>5500). (3) Endothelial cells were removed based on *Pecam1*, *Esam*, *Cd93*, *Cdh5* expression. (4) Epithelial cells were removed based on *Epcam*, *Krt18*, *Krt19* expression. (5) The proliferating cluster included multiple myeloid cell subsets and were removed based on *Mki67* and *Top2a* expression. Number of myeloid cells after quality control was 21249 (10473 and 10776 from the WT and Atg16l1 KO sample respectively). Normalization was implemented with the LogNormalize option in the function NormalizeData. Variable features were identified using the vst method in function FindVariableFeatures. Cluster identification was performed by first constructing a shared nearest neighbor graph in the principal component space, and then optimizing modularity using the Louvain algorithm. UMAP dimensions were used to visualize the resulting clusters. Markers for each cluster were identified by reducing the number of candidate genes to those genes which were (i) at least log(0.25) fold higher expressed in the cluster under consideration compared with all other clusters and (ii) expressed in at least 25% of cells in the cluster under consideration. For genes passing those criteria, significance between cells in the cluster versus all other cells was calculated using Wilcoxon rank sum test and adjusted with the FDR method. Gene set enrichment analysis was performed with the GSEA function in clusterProfiler[78], and the msigdbr function with categories H, C2, and C5. Batch effects between Atg16l1 KO and WT samples were adjusted with the IntegrateData function in Seurat for tumor cells. No batch effects were observed between myeloid cell samples.

### Topic modeling

The raw counts matrix from organoid cells was extracted and filtered to include all cells maintained in the integrated analysis and only the top 10,000 variable features, as determined by FindVariableFeatures with the vst method, as genes. This matrix was used as an input to the FitGoM function in CountClust[79] which fit a topic model to the data using Latent Dirichlet Allocation (LDA). We ran FitGoM with an error tol = 0.1 and $K = 10, 15, 20$, and 25 clusters. The model with $K = 25$ was retained since it best balanced model explanatory power (minimized the Bayesian information criterion of our fit models) with interpretable biological pathways.

Top gene markers and enrichment scores for each gene in a given topic were determined using the ExtractTopFeatures function in CountClust on the theta matrix, containing genes and gene weights for each topic, with method = "poisson", options = "min", and shared =

FALSE. UMAP plots colored by proportional topic assignment in each cell were generated from the omega matrix, containing cells and their topic assignment frequencies (where the sum of topic assignments in a given cell are equal to one).

## RNA velocity analysis

Mouse reference genome and the corresponding gtf file were downloaded from Gencode (GRCm38.primary_assembly.genome.fa.gz and gencode.vM24.annotation.gtf.gz respectively). The eisaR[80] package was used to extract a GRanges object containing the genomic coordinates of each annotated transcript and intron. The 'separate' approach was used to define introns separately for each transcript, and a flank length of 90nt was added to each intron. Reference feature sequences for introns and exons were indexed with Salmon[81], and subsequently quantified with Alevin[82]. Spliced and unspliced counts were imported into R as a SummarizedExperiment object using tximeta[83], converted first into a SingleCellExperiment, and subsequently into a Python Anndata object using zellkonverter (https://github.com/theislab/zellkonverter). scVelo[84] was run on spliced and unspliced counts to estimate RNA velocity with a dynamical model that does not assume a steady-state equilibrium or a common gene-splicing rate (https://scvelo.readthedocs.io/DynamicalModeling/). The connectivity among clusters was estimated using PAGA[85] to infer potential differentiation trajectories.

## IMblaze370 analysis

The IMblaze370 study[33] was performed in accordance with the guidelines for Good Clinical Practice and the Declaration of Helsinki, and the study protocol approval was obtained from independent ethics committees for each participating site. Pre-treatment tumor samples were collected as the paraffin-embedded tissue (FFPE) blocks or sections from patients who gave written informed consent. The pathologic diagnosis of each case was confirmed by review of hematoxylin and eosin (H&E) stained slides and all samples that advanced to nucleic acid extraction contained a minimum of 20% tumor cells. H&E images were marked for macro-dissection by a pathologist. RNA was then extracted from macro-dissected sections using the High Pure FFPET RNA Isolation Kit (Roche). Whole transcriptome profiles were generated using TruSeq RNA Access technology (Illumina®). Strand-specific RNA sequencing was performed and resulting data were analyzed using HTSeqGenie[73] (https://bioconductor.org/packages/HTSeqGenie/) in BioConductor[74] as follows: first, reads with low nucleotide qualities (70% of bases with quality <23) or matches to rRNA and adapter contamination were removed. The reads that passed were then aligned to the human reference genome GRCh38.p10 using GSNAP[75] version '2013-10-10-v2' allowing maximum of two mismatches per 75 base sequence (parameters: '-M 2 -n 10 -B 2 -i 1 -N 1 -w 200000 -E 1 --pairmax-rna=200000'). Transcript annotation was based on the GENCODE 27. To quantify gene expression levels, the number of reads mapping unambiguously to the exons of each gene was calculated. Counts were normalized using the TMM method in the calcNormFactors R package[76]. Median cutoff of normalized *ATG16L1* levels was used to determine high and low levels separately within KRAS mutant and KRAS wildtype tumors. Survival analysis was performed with the survminer R package, and p-values were obtained from median cutoff log-rank tests. Correlation analysis was performed with normalized *ATG16L1* and *EPCAM* levels. Samples were scored for general immune[37], general stroma[37], epithelial[36], and CIBERSORT immune[40] signatures using the mean z-score approach (see 'Computation of signature scores'). To compare *ATG16L1*-high vs low groups, unbiased differential expression, gene set enrichment analysis, and CIBERSORT immune deconvolution were performed with voom+limma[76], fgsea[77] and t-tests, respectively.

## Reporting summary

Further information on research design is available in the Nature Portfolio Reporting Summary linked to this article.

## Data availability

Raw and processed data from in vitro organoid bulk RNA sequencing, and in vivo organoid single cell RNA sequencing have been submitted to Gene Expression Omnibus with accession number GSE192515. Raw data from in vitro organoid whole exome sequencing have been deposited to Sequence Read Archive with accession number PRJNA790973. IMblaze370 bulk RNA sequencing raw data have been submitted to the European Genome-Phenome Archive with accession number EGAS00001005952. Requests for the exploratory biomarker data underlying this publication requires a detailed, hypothesis-driven statistical analysis plan that is collaboratively developed by the requestor and company subject matter experts. Direct such requests to Y.Y. (yan.yibing@gene.com) for consideration. Further details on Roche's Global Policy on the Sharing of Clinical Information and how to request access to related clinical study documents are available online (https://go.roche.com/data_sharing). Anonymized records for individual patients across more than one data source external to Roche cannot, and should not, be linked due to a potential increase in risk of patient re-identification. Publicly available single-cell RNAseq data of human CRC used in this study are available from Gene Expression Omnibus with accession number GSE146771. Publicly available bulk tumor RNAseq data used in this study are available from the TCGA and from Gene Expression Omnibus with accession numbers GSE17536 and GSE39582. The remaining data are available within the Article, Supplementary Information or Source Data file. Source data are provided with this paper.

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

## Acknowledgements

The authors thank A. Gutierrez, G. Ortiz-Munoz and the Vet Staff at Genentech for technical assistance with in vivo studies. We thank the contributions of the Research Pathology Department Core Labs, including multiple contributors from the Center for Advanced Light Microscopy, Necropsy, and Histology labs. We thank R.M. Leitao for assistance with bioinformatic analysis, T. Tran and S. Darmanis at Genentech for assistance with scRNA-seq, A. Gogineni at Genentech for assistance with BLI, S. Marsters at Genentech for providing TRAIL and FASL. We thank Aragen for performing one in vivo study. We thank members of the Murthy and West lab for helpful discussions. Illustrations in Fig. 2a, c, and Supplementary Figs. 2a, 6a, 7a, and 10a, were created with BioRender.com. This work was supported by Genentech.

## Author contributions

L.T. designed and performed experiments, analyzed and interpreted the data, and wrote the manuscript. Y.S. analyzed and interpreted data and wrote the manuscript. L.W. performed experiments and analyzed data. K.B., S.R., A.K.C., D.O., S.J. analyzed data, N.K., J.L., E.M. and M.O. performed experiments. S.G. performed imaging analysis. A.S. and J.Z. performed pathology analysis. G.A., J.B., T.W.K., F.C., M.J.W., and Y.Y. designed and conducted clinical studies, collected patient samples and data. F.J.D.S. and F.D.S.M. provided essential conceptual input. N.R.W. provided essential conceptual input and edited the manuscript. A.M. oversaw the project, interpreted the results and wrote the manuscript.

## Competing interests

Y.S., K.B., N.K., J.L., S.G., L.W., A.S., J.Z., E.M., D.O., S.J, M.W., Y.Y., F.J.D.S, J.B., F.D.S., and N.R.W. are employees of Genentech. L.T. is an employee of Vertex Pharmaceuticals and A.M. is an employee of Gilead Sciences. The remaining authors declare no competing interests.

## Additional information

[1]Department of Cancer Immunology, Genentech Inc., South San Francisco, USA. [2]Department of Oncology Bioinformatics, Genentech Inc., South San Francisco, USA. [3]Department of Molecular Oncology, Genentech Inc., South San Francisco, USA. [4]Center for Advanced Light Microscopy, Genentech Inc., South San Francisco, USA. [5]Department of Pathology, Genentech Inc., South San Francisco, USA. [6]Department of In Vivo Pharmacology, Genentech Inc., South San Francisco, USA. [7]Vall d'Hebrón Institute of Oncology, Vall d'Hebrón University Hospital, Universitat Autònoma de Barcelona, Barcelona, Spain. [8]Sarah Cannon Research Institute/Tennessee Oncology, Nashville, TN, USA. [9]Department of Oncology, Medical Center, University of Ulsan, Seoul, Korea. [10]Department of Precision Medicine, Università degli Studi della Campania Luigi Vanvitelli, Naples, Italy. [11]Oncology Biomarker Development, Genentech, Inc., South San Francisco, CA, USA. [12]Department of Discovery Oncology, Genentech Inc., South San Francisco, USA. [13]Present address: Gilead Sciences, Foster City, USA. [14]These authors contributed equally: Lucia Taraborrelli, Yasin Şenbabaoğlu, Lifen Wang. [15]These authors jointly supervised this work: Nathaniel R. West, Aditya Murthy. ✉e-mail: west.nathaniel@gene.com; aditya.murthy@gilead.com

