## [Peer Review File · Nature Communications]

Tumor-intrinsic expression of the autophagy gene Atg16l1 suppresses anti-tumor immunity in colorectal cancerEditorial Note: This manuscript has been previously reviewed at another journal that is not operating a transparent peer review scheme. This document only contains reviewer comments and rebuttal letters for versions considered at *Nature Communications*.

REVIEWER COMMENTS

Reviewer #1 (Remarks to the Author):

I believe the previous critiques were reasonably addressed and that the manuscript is suitable for publication.

Reviewer #2 (Remarks to the Author):

1. Clinical correlations of ATG16L1 mRNA levels. The additional analyses show weak or absent association of ATG16L1 mRNA levels with survival (DSS or OS). It is curious that the authors include a statement: “We would be willing to include Table R1 as a supplement in the revised manuscript if deemed necessary by the Reviewers”. The authors in fact have an obligation to fully disclose the information to the readers. The analysis should be extended to include all publically available datasets and all core genes in the autophagy pathway. These data should be included in a supplemental table. Furthermore, an explicit statement should be included in the main text. It appears that the conclusion is that the expression level of ATG16L1 is only weakly/not associated with survival in MSS CRC in the absence of immunotherapy. This is consistent with the hypothesis that low ATG16L1 mRNA levels only confer a survival benefit with PD-L1 blockade. Given the lack of a validation dataset from an immunotherapy trial in CRC, this needs to remain a working hypothesis, and this needs to be clearly stated.

2. A related question is whether any other genes in the autophagy pathway show an association with survival in the IMblaze trial. Again, there needs to be a complete analysis on which genes in the pathway show/fail to show an association.

3. Role of T cells/NK cells. This reviewer had requested to show CD8 + NK depletion data in

one figure for WT and KO tumors. The authors now show the data for WT and KO for the CD8 + NK cells depletion group together, but still separated the essential control group (isotype control Ab) into a supplemental figure. The WT/KO comparison and all relevant controls need to be in the same figure (and of course from the same experiment).

4. Was the comparison between WT and Ifng KO mice done in the same experiment to exclude the possibility that experimental variation/stochastic factors impacted the outcome? Specifically, were the experiments in Fig. 2D and 2F done in the same experiment? If not, this analysis needs to be repeated.

Reviewer #3 (Remarks to the Author):

Taraborrelli et al. showed that Atg16l1 which is implicated in autophagy limited cytokine induced tumor cell death and promoted stemness of tumor cells, suggesting that high expression of this gene in cancer maybe associated with resistance to anti-tumor immunity and immunotherapy. Using clinical trial IMblaze 370, they found that high Atg16l1 in KRAS mutated CRC patients associated with lower overall survival. They used mouse APC/Smad4/TRP53 and KRAS G12D colon organoids implanted in immunocompetent WT C57BL/6 or immunocompromised NSG mice to demonstrate the effects of Atg16l1 on adaptive and innate antitumor immunities. Deletion of Atg16l1 increased the sensitivity to IFN and the IFN+TNF induced cell death in culture. Deletion of CTL and NK or deficiency in IFNg decreased the growth of the Atg16l1 KO organoids implanted in WT mice.

Several clarifications and controls have been provided in response to the reviewers included the WT controls in figure 2, as well as knock-out / Knock-in organoids to prove the role of ATG16L1.

The figures and the results are clearly organized, and the rigor of the experimental approach is adequate. The study is of significant interests by proposing that ATG16L1 which is associated with risk of IBD and several GI cancers including CRC, limited the cytokine (IFNg and TNF) induced tumor cell death and antitumor immunity. The authors provided a substantial amount of data (orthotopic mouse model; scRNAseq of epithelial and myeloid cells) and control responding to several experimental approach issues (especially in figure 2 or ATG16L1 knock in organoids) pointed out by reviewers.

However, the overall impression still remains that the study is incomplete and “patchy” with accumulation of extrinsic and extrinsic effects linked to ATG16L1 deletion but without real connection to better understand the mechanism of action of ATG16L1. Several of the sections are descriptive (including the EMT effect on organoids or the myeloid effect in the NSG mice). One the remaining major issue is that understanding if deletion of ATG16L1 impacted the TME, using NSG mice, even to the limited study of the myeloid compartment, is not adequate. In absence of the T cell compartment the myeloid cells are behaving very differently and may not reflect the reality. Never-the-less, NSG mice remain ok to study tumor intrinsic effects of ATG16L1 deletion. Noteworthy, looking at fig 2, it seems that there is enough variability in tumor growth of ATG16L1 KO organoid growth in WT BL6 mice to be able to get some remaining tumors and evaluate the TME.

It is unclear how to put together the immune effects described in fig2d, e, f, g and the data in the NSG mice to get a mechanistic explanation of the effect of ATG16L1 deletion on the immune response. For example, it would be interesting to test the immunogenicity of cytokine-induced cell death by implanting ATG16L1 KO organoids in immunocompetent mice and re-challenging with WT organoid to see if the growth of the organoid is indeed limited (=immune memory; anti-tumor immunity further tested by CD8 and NK depletion). The authors should also dissociate the effects of NK and CD8 T cells to understand how these immune effector cells interact and contribute to the anti-tumor immunity. Finally, the authors proposed that ATG16L1 high may predict resistance to immunotherapy; but the overall manuscript is not testing the immunotherapy and there is not enough direct evidence that ATG16L1 impacts the antitumor immunity as claimed by the authors.

Additional comment:

Line 405 or line 450: the authors can't claim, in the state of this study, that ATG16L1 expression is a biomarker of poor response to the PD-L1 inhibitor atezolizumab in patient with mutated KRAS (no power and validation cohort). At the most, the study will warrant further investigation to test if ATG16L1 could be considered as a biomarker of poor response to immune checkpoint blockade. The experimental approach did not explore this.

Reviewer #4 (Remarks to the Author): new referee

In this study, Taraborrelli et al. revealed the molecular mechanisms through which high expression of ATG16L1 negatively impacts the effectiveness of immunotherapy for advanced metastatic colorectal cancer (CRC) with KRAS mutations in the clinical trial IMblaze 370. The authors found that deletion of ATG16L1 prevents tumor growth in immunocompetent mice, but its effectiveness decreases with the severity of immune deficiency in different mice models. This finding is intriguing and novel. The authors established the dependence of the interaction between tumor cells and the immune system on ATG16L1 deletion in mice models and showed ATG16L1 in epithelial cells suppresses cellular immunity through inhibiting TNF and IFN γ -induced necroptosis and apoptosis, likely via RIPK3. They presented a thorough set of experiments to support their conclusions.

From a bioinformatics perspective, the methods used in this study were thorough and appropriate. The authors utilized bulk gene expression data from IMblaze 370 and GSE17536 and performed CIBERSORTx deconvolution analysis to circumvent confounding and mixed effects inherent in bulk data and to determine the correlation of ATG16L1 expression with immune and epithelial signatures. They analyzed scRNA-seq dataset GSE146771 and found higher ATG16L1 expression in tumor epithelium compared to other cell types in the CRC TIME. However, since ATG16L1 expression was also observed in T cells (Fig. 1d, e), it is necessary to consider the potential confounding effects on the analysis and the authors should comment on the non-specific expression of ATG16L1 in other cells, especially T cells, that may impact anti-tumor immunity.

The authors also conducted scRNA-seq experiments to understand the impact of ATG16L1 on tumor and microenvironment phenotypic programming by profiling organoids implanted in wild-type and ATG16L1 knock-out NSG mice. They used the widely-accepted scRNA-seq analysis package Seurat and implemented downstream methods like Gene Set Enrichment Analysis and RNA velocity. Quality control steps were performed separately for tumor and immune cells, with normalization using appropriate parameters. The methods used in the analysis were standard and appropriate, with reasonable parameter settings.

The overall quality of the manuscript is high, with clear, concise writing and well-designed

experiments. The conclusions reached by the authors are supported by the data presented in a thorough and convincing manner. There are, however, some minor issues that should be corrected in a revised version. Specifically, it was not immediately clear regarding to the number of samples used for scRNA-seq (is it 2 knockout vs. 2 wild-type mice?) (Fig. 3 & 4). Furthermore, the statistical methods used in Figures 3e, 3g and 4e were not described.

These minor points aside, this study provides a novel finding, establishing ATG16L1 expression as a predictive biomarker for poor response to PD-L1 inhibitor atezolizumab MSS CRC patients with KRAS mutations. The authors' conclusions provide a rational basis for the inhibition of autophagy as a means to enhance immunotherapy responses in the clinical setting.

Response to Reviewers, NCOMMS-23-04608-T

Reviewer #1 (Remarks to the Author):

I believe the previous critiques were reasonably addressed and that the manuscript is suitable for publication.

We thank the Reviewer for acknowledging our revision and suitability of the manuscript for publication.

Reviewer #2 (Remarks to the Author):

1. Clinical correlations of ATG16L1 mRNA levels. The additional analyses show weak or absent association of ATG16L1 mRNA levels with survival (DSS or OS). It is curious that the authors include a statement: "We would be willing to include Table R1 as a supplement in the revised manuscript if deemed necessary by the Reviewers". The authors in fact have an obligation to fully disclose the information to the readers. The analysis should be extended to include all publically available datasets and all core genes in the autophagy pathway. These data should be included in a supplemental table. Furthermore, an explicit statement should be included in the main text.

We thank the Reviewer for this suggestion. In our previous revision, we provided analysis of outcome data for all core autophagosome elongation genes (ATG3, ATG7, ATG10, ATG5, ATG12, ATG4A, ATG4B, ATG4C, ATG4D) in 3 studies (GSE17536, GSE39582, TCGA). These data are now provided in the revised manuscript as Extended Data Table 2. We also assessed a number of additional studies (GSE39084, GSE17537, GSE33113, GSE24551, GSE13067, GSE13294, GSE18088, GSE26682, GSE41258, and GSE14333). Our bioinformatics co-authors decided to omit these additional studies in final analysis due to insufficient numbers of patients with metastatic disease ($n < 25$). As per the Reviewer's guidance, we have detailed these findings in the revised manuscript (Page 4):

Given that *IMBlaze 370* remains the only well-powered clinical investigation for immunotherapy in late stage MSS-CRC, additional analysis was limited to observational studies with significantly smaller cohort sizes of stage IV disease (GSE17536, GSE39582, TCGA; Extended Data Table 2). Immunotherapy was not included as a treatment arm in these studies, and subcohorts of non-MSI patients harboring KRAS mutations further decreased patient numbers. With these caveats, a poor prognostic association was observed for *ATG16L1* in GSE17536, and for *ATG7*, *ATG10*, and *ATG4A* in GSE39582. Small patient numbers ($n < 25$) precluded analysis of a number of additional datasets (GSE39084, GSE17537, GSE33113, GSE24551, GSE13067, GSE13294, GSE18088, GSE26682, GSE41258, and GSE14333). These analyses suggest a weak or absent association between *ATG16L1* transcript levels and patient outcome in non-immunotherapy settings. Caution must be exercised when considering these studies due to small cohort sizes and lack of appropriate treatment arms.

Our gene set includes the core components of the autophagosome elongation complex, since these are specific essential components of autophagosome formation (illustrated in Extended Data Figure 2a in the revision). Members of the upstream pathways involved in autophagy are also essential for non-autophagic processes, thus an

autophagy-specific contribution of these genes cannot be assumed (reviewed in Ref. 17 and 18 of the revision). Murine knockout models corroborate these findings as germline deletion of core autophagy genes confer perinatal lethality while lack of the upstream genes result in embryonic lethality, revealing additional pathways governed by these genes. This context has been provided in the Discussion (Page 11). We thank the Reviewer for prompting the inclusion of this relevant context:

Several actionable targets within the autophagic flux program make it an attractive pathway for therapeutic modulation; however, caution needs to be exercised when considering regulators of membrane trafficking (e.g., VPS34, Unc51-like kinases ULK1/2) or lysosomal fitness (e.g., CQ/HCQ, other lysosomal inhibitors). These represent master regulatory nodes that impact multiple processes beyond autophagy, and inhibitors of these targets may thus be limited by toxicity. Germline knockout models provide further evidence of pathway divergence, where loss of autophagosome elongation genes *Atg3*, *5*, *7*, *10*, *12* and *Atg16l1* demonstrate neonatal lethality, while deletion of *Beclin 1*, *Vps34*, *Rb1cc1* result in embryonic lethality. Single deletion of *ULK1* or *ULK2* appears to be tolerated, however combined deletion results in neonatal lethality associated with both autophagy-dependent and independent processes⁶⁶⁻⁶⁹. Sustained pharmacological inhibition of *VPS34* was shown to be poorly tolerated, consistent with its essential role in mammalian development⁷⁰. Finally, while multiple clinical trials are currently investigating CQ/HCQ in cancer, there lacks mechanistic demonstration that any therapeutic benefit conferred by these lysosomal modulators is primarily due to autophagy inhibition. These findings highlight the need to focus on more specific modulators of autophagosome formation, for example members of the *ATG16L1* complex and the de-ubiquitinases of the *ATG4* family which recycle LC3 to sustain autophagic flux.

It appears that the conclusion is that the expression level of *ATG16L1* is only weakly/not associated with survival in MSS CRC in the absence of immunotherapy. This is consistent with the hypothesis that low *ATG16L1* mRNA levels only confer a survival benefit with PD-L1 blockade. Given the lack of a validation dataset from an immunotherapy trial in CRC, this needs to remain a working hypothesis, and this needs to be clearly stated.

We acknowledge the absence of an appropriate validation dataset for immunotherapy in MSS-CRC. We have explicitly stated this in the revised Discussion (Page 10):

A lack of independent clinical datasets to validate these encouraging observations presents a limitation to the study. We acknowledge that the data must be considered as an initial finding, warranting confirmation in follow-up studies with comparable patient populations and treatment arms.

2. A related question is whether any other genes in the autophagy pathway show an association with survival in the IMBlaze trial. Again, there needs to be a complete analysis on which genes in the pathway show/fail to show an association.

Building on the above query, we asked whether *ATG3*, *ATG4A-D*, *ATG5*, *ATG7*, *ATG10* and *ATG12* transcript levels are also enriched in the tumor epithelium as observed for *ATG16L1*, and whether they similarly associate with patient outcome in *IMBlaze 370*. Among these genes, only *ATG16L1* appears to be enriched in the tumor epithelium

compared to other components of the CRC tissue microenvironment (Extended Data Figure 2b in the revision). Consistently, we do not find the other genes to specifically associate with patient outcome in both immunotherapy treatment arms (Atezolizumab monotherapy and Atezolizumab + Cobimetinib) in the KRAS mutant subcohort. These new findings have been discussed in the revised manuscript (Page 4):

Expanded analysis of core components of the autophagosome elongation machinery (illustrated in Extended Data Fig. 2a) showed *ATG16L1* to be preferentially enriched in the tumor epithelium when compared to *ATG3*, *ATG4B*, *ATG5*, *ATG7*, *ATG10* and *ATG12*. *ATG4A*, *ATG4C* and *ATG4D* were not detected, likely due to poor transcript coverage or low expression (Extended Data Fig. 2b). Additionally, only *ATG16L1* showed a significant association with poor outcome under immunotherapy regimens in KRAS-mutant disease in *IMblaze 370* (Atezolizumab monotherapy, Atezolizumab + Cobimetinib; Extended Data Table 1).

3. Role of T cells/NK cells. This reviewer had requested to show CD8 + NK depletion data in one figure for WT and KO tumors. The authors now show the data for WT and KO for the CD8 + NK cells depletion group together, but still separated the essential control group (isotype control Ab) into a supplemental figure. The WT/KO comparison and all relevant controls need to be in the same figure (and of course from the same experiment).

The WT/KO comparisons are indeed from the same experiment, and we agree that providing all the tumor growth data in a singular plot will explicitly convey this fact. We have improved on the depletion studies by generating individual NK and CD8+ T cell depletion in new models, along with relevant controls. Remarkably, we observe that depletion of NK cells markedly rescues growth of *ATG16L1* KO CRC organoids, whereas depletion of CD8+ T cells has a comparatively modest impact. These findings are shown in Fig. 2f, Fig. 2h, Extended Data Fig. 8c and described in the revised manuscript (Page 6):

...we individually depleted host cytotoxic T lymphocytes (CTL, CD8+ T cells) or natural killer (NK) cells followed by implantation of CRC organoids. Near complete, sustained loss of CD8+ T cells or NK cells was observed 4 weeks following administration of depleting antibodies compared to non-depleting isotype controls (Extended Data Fig. 8a, b). Interestingly, depletion of NK cells markedly rescued growth of *ATG16L1* KO CRC organoids, while depletion of CD8+ T cells had a comparatively modest impact on tumor growth (Fig. 2f; Fig 2h, WT host vs. CD8 or NK depleted hosts; tumor growth curves in Extended Data Fig. 8c). Thus, NK cells seem to be key contributors to the clearance of *ATG16L1*-deficient MSS-CRC.

4. Was the comparison between WT and Ifng KO mice done in the same experiment to exclude the possibility that experimental variation/stochastic factors impacted the outcome? Specifically, were the experiments in Fig. 2D and 2F done in the same experiment? If not, this analysis needs to be repeated.

The experiments in Fig. 2D and 2F of the previous version were performed separately in WT and IFNg KO mice due to lack of mouse availability within a single study. We included the same WT CRC organoid clone across studies to enable cross-study

comparisons. In the latest revision, we were able to obtain sufficiently sized cohorts of WT and IFN γ KO mice to generate all experimental groups (WT tumor, WT host; KO tumor, WT host; WT tumor, KO host; KO tumor, KO host) within the same study. As shown below, we confirm our earlier findings that loss of host IFN γ significantly increases liver colonization by ATG16L1 KO CRC organoids.

Confirmation of IFN γ KO phenotype. WT or IFN γ KO C57BL/6 mice were implanted in the liver with either WT or Atg16l1 organoids by HTV (n=10 per group). Tumor growth was monitored over time by BLI. **(a)** Week 5 BLI signal; fold differences between WT and Atg16l1 KO groups are shown separately for WT or IFN γ KO hosts. **(b)** Longitudinal BLI data from the same experiment as panel a. In this plot, week 5-fold differences between WT and IFN γ KO host mice are shown separately for groups that received either WT or Atg16l1 KO organoids. Fold differences were calculated based on group medians. Statistical significance determined using Mann-Whitney test; **P<0.01, ***P<0.001, ****P<0.0001.

Reviewer #3 (Remarks to the Author):

Taraborrelli et al. showed that Atg16l1 which is implicated in autophagy limited cytokine induced tumor cell death and promoted stemness of tumor cells, suggesting that high expression of this gene in cancer maybe associated with resistance to anti-tumor immunity and immunotherapy. Using clinical trial IMblaze 370, they found that high Atg16l1 in KRAS mutated CRC patients associated with lower overall survival. They used mouse APC/Smad4/TRP53 and KRAS G12D colon organoids implanted in immunocompetent WT C57BL/6 or immunocompromised NSG mice to demonstrate the effects of Atg16l1 on adaptive and innate antitumor immunities. Deletion of Atg16l1 increased the sensitivity to IFN and the IFN+TNF induced cell death in culture. Deletion of CTL and NK or deficiency in IFN γ decreased the growth of the Atg16l1 KO organoids implanted in WT mice.

Several clarifications and controls have been provided in response to the reviewers included the WT controls in figure 2, as well as knock-out / Knock-in organoids to prove the role of ATG16L1. The figures and the results are clearly organized, and the rigor of the experimental approach is adequate. The study is of significant interests by proposing that ATG16L1 which is associated with risk of IBD and several GI cancers including CRC, limited the cytokine (IFN γ and TNF) induced tumor cell death and antitumor immunity. The authors provided a substantial amount of data (orthotopic mouse model; scRNAseq of epithelial and myeloid cells) and control responding to several experimental approach issues (especially in figure 2 or ATG16l1 knock in organoids) pointed out by reviewers.

We thank the Reviewer for acknowledging that our revisions addressed their initial queries, and endorsing the significance of the study.

However, the overall impression still remains that the study is incomplete and “patchy” with accumulation of extrinsic and extrinsic effects linked to ATG16L1 deletion but without real connection to better understand the mechanism of action of ATG16L1. Several of the sections are descriptive (including the EMT effect on organoids or the myeloid effect in the NSG mice). One of the remaining major issues is that understanding if deletion of ATG16L1 impacted the TME, using NSG mice, even to the limited study of the myeloid compartment, is not adequate. In the absence of the T cell compartment the myeloid cells are behaving very differently and may not reflect the reality. Nevertheless, NSG mice remain ok to study tumor intrinsic effects of ATG16L1 deletion. Noteworthy, looking at fig 2, it seems that there is enough variability in tumor growth of ATG16L1 KO organoid growth in WT BL6 mice to be able to get some remaining tumors and evaluate the TME.

Our analysis of KO organoid growth in WT BL6 mice clearly demonstrates drastically compromised tumor growth *in vivo*. The variability in BLI signal strength of KO organoids in Fig. 2d arises from 3 data points with detectable BLI signal (from a total of 11 mice). However, gross and histological assessment of the liver revealed that these mice did not contain sufficient tumor burden to enable meaningful analysis of the immune infiltrate (Figure 2i, Extended data 4b, d, e). Attempts to harvest and analyze this minimal tumor burden were unsuccessful. More importantly, we are concerned that any gene expression analyses of these “outlier” tissues would be potentially irreproducible and provide non-interpretable data for the reader. Analysis of spurious outgrowths will likely not provide additional insight for the role of tumor-intrinsic ATG16L1 in anti-tumor immunity. We hope you agree with our concern regarding this request. To provide further clarity, we now depict individual data points in all our tumor growth (BLI) plots illustrating the true spread of data within each group.

It is unclear how to put together the immune effects described in fig2d, e, f, g and the data in the NSG mice to get a mechanistic explanation of the effect of ATG16L1 deletion on the immune response. For example, it would be interesting to test the immunogenicity of cytokine-induced cell death by implanting ATG16L1 KO organoids in immunocompetent mice and re-challenging with WT organoid to see if the growth of the organoid is indeed limited (=immune memory; anti-tumor immunity further tested by CD8 and NK depletion). The authors should also dissociate the effects of NK and CD8 T cells to understand how these immune effector cells interact and contribute to the anti-tumor immunity.

We thank the Reviewer for these suggestions. We have performed individual depletion of NK or CD8+ T cells to obtain a better understanding of their contributions to anti-tumor immunity in our models. We show that depletion of NK cells markedly rescued growth of ATG16L1-deficient CRC organoids. In contrast, depletion of CD8+ T cells had a much more modest effect. This is consistent with the lack of antigen-specific immunogenicity in MSS-CRC and the acknowledged association of tumor mutation/MSI-status with immunotherapy responses in the clinic⁸. These new findings are reported in Fig. 2f, 2h and Extended Data Fig. 8a-c of the revised manuscript (Results, Page 6):

...we individually depleted host cytotoxic T lymphocytes (CTL, CD8+ T cells) or natural killer (NK) cells followed by implantation of CRC organoids. Near complete, sustained loss of CD8+ T cells or NK cells was observed 4 weeks following administration of depleting antibodies compared to non-depleting isotype controls (Extended Data Fig. 8a, b). Interestingly, depletion of NK cells markedly rescued growth of ATG16L1 KO CRC organoids, while depletion of CD8+ T cells had a comparatively modest impact on tumor growth (Fig. 2f; Fig

2h, WT host vs. CD8 or NK depleted hosts; tumor growth curves in Extended Data Fig. 8c). Thus, NK cells seem to be key contributors to the clearance of ATG16L1-deficient MSS-CRC.

In contrast to MSI-CRC which exhibits a high degree of tumor mutation burden and potential for neoantigen generation, MSS-CRC is widely acknowledged as immunologically 'cold', and there is strong consensus that there is little to no antigen-specific immunogenicity in this setting¹⁻³. In our models, deficiency of IFN γ or NK cells in hosts significantly rescued growth of ATG16L1 KO CRC organoids, suggesting a primarily antigen-independent mechanism of action. This is consistent with the lack of antigen-specific immunity observed in MSS-CRC, and directly addresses the above Reviewer query regarding the role of tumor antigen-specific T cells in our study. We discuss this in the revised manuscript (Discussion, Page 11) in light of the new findings:

Intriguingly, we observed that NK cells and IFN γ , but not CD8+ T cells, were primary drivers of cytotoxicity against ATG16L1-deficient tumors. This posits an antigen-independent mechanism of anti-tumor immunity in our model, and is consistent with a lack of tumor mutation burden (TMB) in MSS-CRC, a basis for generally poor immunotherapy responses in CRC^{64,65}. Our findings thus expand the scope of autophagy modulation as a therapeutic avenue in MSS-CRC.

We would like to thank the Reviewer for proposing these studies, as they have meaningfully improved our manuscript and provided significant clarity to the contribution of cytotoxic T lymphocytes or natural killer cells and suppression of MSS-CRC.

Finally, the authors proposed that ATG16L1 high may predict resistance to immunotherapy; but the overall manuscript is not testing the immunotherapy and there is not enough direct evidence that ATG16L1 impacts the antitumor immunity as claimed by the authors.

Additional comment:

Line 405 or line 450: the authors can't claim, in the state of this study, that ATG16L1 expression is a biomarker of poor response to the PD-L1 inhibitor atezolizumab in patient with mutated KRAS (no power and validation cohort). At the most, the study will warrant further investigation to test if ATG16L1 could be considered as a biomarker of poor response to immune checkpoint blockade. The experimental approach did not explore this.

We would like to reiterate that *IMBlaze370* remains the *only* Phase III trial assessing checkpoint blockade in MSS-CRC for which RNAseq data are available. Our work discloses the clinical outcomes associated with ATG16L1 and provides the first analysis of patient gene expression profiles in a unique clinical setting. *IMBlaze370* is a well-powered multi-cohort study, with **n=363** across all treatment groups and **n=140** for the relevant comparisons (Atezolizumab monotherapy, Atezolizumab+Cobimetinb in KRAS mutant patients). None of the analyzed publicly available datasets, including TCGA, achieve these patient numbers for Stage IV metastatic MSS-CRC (Extended Data Table 2 in the revision). Thus, while we have explicitly stated in the Discussion that our findings warrant further validation due to lack of an independent Phase III dataset, we maintain that this is a novel finding based on a highly significant clinical association and respectfully disagree with the Reviewer that the clinical study lacks power.

The drastically reduced growth of ATG16L1 KO tumors in multiple *in vivo* settings precluded additional comparison of immunotherapy by checkpoint blockade, as no

additional therapeutic benefit could be achieved. Assessment of potential combinations with immunotherapy are the focus of follow-up studies with more sophisticated genetic and pharmacological tools, and remain out of scope for this current manuscript. We maintain, and hope the Reviewer agrees, that the clear pre-clinical phenotypes, combined with the significant association of *ATG16L1* with patient outcome specifically in context of Atezolizumab treatment in *IMBlaze 370* together support our conclusion that *ATG16L1* contributes to a resistance to anti-tumor immunity.

Reviewer #4 (Remarks to the Author): new referee

In this study, Taraborrelli et al. revealed the molecular mechanisms through which high expression of *ATG16L1* negatively impacts the effectiveness of immunotherapy for advanced metastatic colorectal cancer (CRC) with *KRAS* mutations in the clinical trial *IMBlaze 370*. The authors found that deletion of *ATG16L1* prevents tumor growth in immunocompetent mice, but its effectiveness decreases with the severity of immune deficiency in different mice models. This finding is intriguing and novel. The authors established the dependence of the interaction between tumor cells and the immune system on *ATG16L1* deletion in mice models and showed *ATG16L1* in epithelial cells suppresses cellular immunity through inhibiting TNF and IFN γ -induced necroptosis and apoptosis, likely via RIPK3. They presented a thorough set of experiments to support their conclusions.

From a bioinformatics perspective, the methods used in this study were thorough and appropriate. The authors utilized bulk gene expression data from *IMBlaze 370* and GSE17536 and performed CIBERSORTx deconvolution analysis to circumvent confounding and mixed effects inherent in bulk data and to determine the correlation of *ATG16L1* expression with immune and epithelial signatures. They analyzed scRNA-seq dataset GSE146771 and found higher *ATG16L1* expression in tumor epithelium compared to other cell types in the CRC TIME. However, since *ATG16L1* expression was also observed in T cells (Fig. 1d, e), it is necessary to consider the potential confounding effects on the analysis and the authors should comment on the non-specific expression of *ATG16L1* in other cells, especially T cells, that may impact anti-tumor immunity.

We thank the reviewer for this careful evaluation of our bioinformatics analyses and agree that despite its enrichment in tumor epithelial cells, non-specific expression of *ATG16L1* in T cells could confound our analysis, particularly since T cells are generally expected to associate with more favorable immunotherapy outcome (an effect opposite to that of *ATG16L1*). To acknowledge this point, we have included the following statement in the Discussion of the revised manuscript (page 10):

Consistently, *ATG16L1* exhibited a clear enrichment in CRC epithelium when compared with the above *ATG* genes. Nevertheless, detectable expression of *ATG16L1* in non-epithelial cells (particularly T cells) from a minority of samples could confound the interpretation of bulk tissue transcriptome data, and future histopathological studies of *ATG16L1* protein expression in large patient cohorts should further clarify the association of epithelial *ATG16L1* expression with immunotherapy outcome.

The authors also conducted scRNA-seq experiments to understand the impact of *ATG16L1* on tumor and microenvironment phenotypic programming by profiling organoids implanted in wild-

type and ATG16L1 knock-out NSG mice. They used the widely-accepted scRNA-seq analysis package Seurat and implemented downstream methods like Gene Set Enrichment Analysis and RNA velocity. Quality control steps were performed separately for tumor and immune cells, with normalization using appropriate parameters. The methods used in the analysis were standard and appropriate, with reasonable parameter settings.

The overall quality of the manuscript is high, with clear, concise writing and well-designed experiments. The conclusions reached by the authors are supported by the data presented in a thorough and convincing manner. There are, however, some minor issues that should be corrected in a revised version. Specifically, it was not immediately clear regarding to the number of samples used for scRNA-seq (is it 2 knockout vs. 2 wild-type mice?) (Fig. 3 & 4). Furthermore, the statistical methods used in Figures 3e, 3g and 4e were not described.

These minor points aside, this study provides a novel finding, establishing ATG16L1 expression as a predictive biomarker for poor response to PD-L1 inhibitor atezolizumab MSS CRC patients with KRAS mutations. The authors' conclusions provide a rational basis for the inhibition of autophagy as a means to enhance immunotherapy responses in the clinical setting.

We thank the Reviewer for their comments. We have provided the sample number directly in the Figure Legend for each dataset in question. Cells were harvested from n=2 mice for each condition (2 knockout, 2 wild-type).

Additional description of statistical methods is also provided directly in the respective figure legend:

Fig. 3e: Significance levels were derived from adjusted p-values with p-values from Pearson's chi-squared test for two proportions as implemented in the prop.test R function, and adjusted values from false discovery rate correction.

Fig. 3g: fgsea with voom+limma derived fold changes, unadjusted p values shown

Fig. 4e: Significance levels were derived from adjusted p-values with p-values from Pearson's chi-squared test for two proportions as implemented in the prop.test R function, and adjusted values from false discovery rate correction.

References

1. Overman, M. J., Ernstoff, M. S. & Morse, M. A. Where We Stand With Immunotherapy in Colorectal Cancer: Deficient Mismatch Repair, Proficient Mismatch Repair, and Toxicity Management. *Am. Soc. Clin. Oncol. Educ. Book* **38**, 239–247 (2018).
2. Quintanilha, J. C. F. *et al.* Comparative Effectiveness of Immune Checkpoint Inhibitors vs Chemotherapy in Patients With Metastatic Colorectal Cancer With Measures of Microsatellite Instability, Mismatch Repair, or Tumor Mutational Burden. *JAMA Netw. Open* **6**, e2252244 (2023).
3. Diaz, L. A. *et al.* Pembrolizumab versus chemotherapy for microsatellite instability-high or mismatch repair-deficient metastatic colorectal cancer (KEYNOTE-177): final analysis of a randomised, open-label, phase 3 study. *Lancet Oncol.* **23**, 659–670 (2022).

REVIEWERS' COMMENTS

Reviewer #2 (Remarks to the Author):

The authors have addressed most of my questions. I have two smaller issues: 1. The figure in response to question #4 (labeled as 'Confirmation of IFN γ KO phenotype') should be included in the supplement and not only in the rebuttal letter. 2. In Fig. 2F, statistical comparisons should also be made across antibody treatment groups for Atg16l1 KO organoids (in particular for NK cell depletion versus isotype control Ab groups) because a conclusion is made that NK cells are relevant in immunity to Atg16l1 KO tumors. The same applies to all Atg16l1 KO groups in Fig. 2H.

Reviewer #3 (Remarks to the Author):

The authors have answered the best they could to the reviewer comments. Nevertheless, the manuscript still suffers a lack of mechanisms linking ATG16L1 in epithelial cells with NK activation and killing (NK-derived IFN γ ? NK activation molecules? MHC-I expression molecules)

Several comments should be addressed to clarify the data analysis.

A-Statistics

Figure 2H. Are the stats missing or not significant? The non significant p should be indicated (n.s) so that it is clear that the statistics have been done.

Fig.5 D and F. the statistic are missing

As well as in Extended data 1, 5a, 8a and b

B-The flow cytometry dot plots in extended data 8b are not readable

C- Result:

Figure 2E and 2J. The results shown with the stats (significant p values) in Fig.2E are not clear compared to the picture in 2J (no obvious difference between WT and ATG16L1KO) and the comments in the RESULTS section (line 226-248). The authors should clarified what these results in NSG mice mean.

Regarding the effects of CD8 and NK depletion experiments in ATG16L1 KO, the authors commented in their rebuttal that CD8 depletion showed a modest effect on tumor growth. However, figure 2 showed no effect at all at least in the liver.

Regarding the description of the results of fig.5F in the RESULTS section line 395 :” Deletion of Ripk3 partially rescued TNF+IFN γ -induced death in Atg16l1 KO CRC organoids, and cell death was completely blocked by addition of z-VAD to cells doubly deficient in ATG16L1 and RIPK3 (Fig. 5f). Caspase inhibition in ATG16L1 KO organoids further accelerated cell death, likely via RIPK3 mediated necroptosis, consistent with a regulatory role of caspases in necroptosis.” Is that what the figure 5f really shows? It seems that the deletion of RIPK3 in ATG6L1 KO CRC organoids partially rescued the survival of the organoids upon the TNF+IFN treatment and do not rescue the cell death and addition of zVAD completely inhibit cell death. If this is the case, the sentence may be reworded to reflect the data in the figure.

Reviewer #4 (Remarks to the Author):

The authors have addressed all concerns and I do not have any more questions.

Response to Reviewers

August 31, 2023

*Taraborelli, Senbabaoglu, Wang et al. **Tumor-intrinsic expression of the autophagy gene Atg16l1 suppresses anti-tumor immunity in colorectal cancer***

We thank the Reviewers for evaluating our manuscript once more, and for continuing to provide constructive feedback to improve the interpretability of our data. Our point-by-point responses to the latest Reviewer comments can be found below.

Reviewer #2 (Remarks to the Author):

The authors have addressed most of my questions. I have two smaller issues:

1. The figure in response to question #4 (labeled as 'Confirmation of IFN γ KO phenotype') should be included in the supplement and not only in the rebuttal letter.

The longitudinal BLI data associated with this experiment are now included as **Supplementary Fig. 8d** in the revised manuscript and the corresponding Results section has also been updated accordingly.

2. In Fig. 2F, statistical comparisons should also be made across antibody treatment groups for Atg16l1 KO organoids (in particular for NK cell depletion versus isotype control Ab groups) because a conclusion is made that NK cells are relevant in immunity to Atg16l1 KO tumors. The same applies to all Atg16l1 KO groups in Fig. 2H.

To emphasize the impact of T cell and NK cell depletion, Fig. 2f has been re-organized and includes new statistics, as suggested by the reviewer. The original statistical comparisons are also included. Fig. 2h has also been updated to include statistical comparisons of Atg16l1 KO organoids from each condition with Atg16l1 KO organoid growth in WT/untreated control mice.

Reviewer #3 (Remarks to the Author):

The authors have answered the best they could to the reviewer comments. Nevertheless, the manuscript still suffers a lack of mechanisms linking ATG16L1 in epithelial cells with NK activation and killing (NK-derived IFN γ ? NK activation molecules? MHC-I expression molecules) Several comments should be addressed to clarify the data analysis.

A-Statistics

Figure 2H. Are the stats missing or not significant? The non significant p should be indicated (n.s) so that it is clear that the statistics have been done.

Statistics have been added to Fig. 2H as suggested.

Fig.5 D and F. the statistic are missing
As well as in Extended data 1, 5a, 8a and b

To Fig. 5D and 5F, we have added statistical analysis (ANOVA) to compare relevant groups.

Statistical comparisons are provided for all panels of Supplementary Fig. 1 (previously Extended Data Fig. 1).

Statistical comparisons (multiple testing-corrected Mann-Whitney tests) are now included for Supplementary Fig. 5a (previously Extended Data Fig. 5a).

Groups in Supplementary Fig. 8a/b (previously Extended Data Fig. 8a/b) are now compared using multiple-testing corrected ANOVA.

B-The flow cytometry dot plots in extended data 8b are not readable

We apologize for the inconvenience and have used a higher resolution figure for the revised Supplementary Data.

C- Result:

Figure 2E and 2J. The results shown with the stats (significant p values) in Fig.2E are not clear compared to the picture in 2J (no obvious difference between WT and ATG1611KO) and the comments in the RESULTS section (line 226-248). The authors should clarified what these results in NSG mice mean.

We have stated the following in the results section:

“Although tumor burden in the Atg1611 KO group remained significantly decreased compared to the WT group (approximately a 2-fold difference in BLI signal after 4 weeks), all NSG mice administered with Atg1611 KO CRC organoids presented with large tumor nodules (Fig. 2j).”

The data shown in Fig. 2e are consistent with our description in the Results; although the difference in tumor BLI signal between WT and Atg1611 KO organoids is relatively modest in NSG mice (~2-fold), it is nevertheless statistically significant, as indicated in the figure.

Regarding the effects of CD8 and NK depletion experiments in ATG16L1 KO, the authors commented in their rebuttal that CD8 depletion showed a modest effect on tumor growth. However, figure 2 showed no effect at all at least in the liver.

We thank the reviewer for raising this point and have clarified the relevant portion of the Results section to state that we observed no significant effect of CD8 depletion on tumor growth.

Regarding the description of the results of fig.5F in the RESULTS section line 395 :” Deletion of Ripk3 partially rescued TNF+IFN γ -induced death in Atg16l1 KO CRC organoids, and cell death was completely blocked by addition of z-VAD to cells doubly deficient in ATG16L1 and RIPK3 (Fig. 5f). Caspase inhibition in ATG16L1 KO organoids further accelerated cell death, likely via RIPK3 mediated necroptosis, consistent with a regulatory role of caspases in necroptosis.” Is that what the figure 5f really shows? It seems that the deletion of RIPK3 in ATG6L1 KO CRC organoids partially rescued the survival of the organoids upon the TNF+IFN treatment and do not rescue the cell death and addition of zVAD completely inhibit cell death. If this is the case, the sentence may be reworded to reflect the data in the figure.

We have re-examined the data and our description of the findings in the Results section, and feel that our description is accurate. This cell death assay involves detection of the membrane-impermeable DNA intercalating agent propidium iodide (PI) in dead/dying cells, and therefore increasing values on the y-axis indicate increasing cell death. Thus it is clear that under conditions of TNF+IFN γ treatment, Atg16l1/RIPK3 dKO cells (solid black points) show less death than Atg16l1 KO cells (solid red points). Similarly, addition of the caspase inhibitor zVAD further reduces cell death (nearly to the level of untreated cells) in cytokine-treated Atg16l1/RIPK3 dKO cells (gray points). In Atg16l1 single KO cells treated with TNF+IFN γ , zVAD increases cell death (gold points), likely due to disruption of the regulatory effect of caspases on necroptosis. Inhibition of caspases using zVAD is known to promote cell death via necroptosis, and we and others have shown that ATG16L1 loss promotes necroptosis under such conditions (Lim et al, eLife, 2019; Matsuzawa-Ishimoto et al, J Exp Med, 2017). Thus, we conclude that TNF+IFN γ -induced death of Atg16l1-deficient CRC organoids is mediated by both caspases and necroptosis.

Reviewer #4 (Remarks to the Author):

The authors have addressed all concerns and I do not have any more questions.

We thank the reviewer for evaluating our manuscript once more, and are glad to have addressed his/her comments satisfactorily.